# Nance-Horan-syndrome-like 1b controls mesodermal cell migration by regulating protrusion and actin dynamics during zebrafish gastrulation

Sophie Escot [1] ✉, Yara Hassanein[1], Amélie Elouin [1], Jorge Torres-Paz[2], Lucille Mellottee[1], Amandine Ignace[1] & Nicolas B. David [1] ✉

Cell migrations are crucial for embryonic development, wound healing, the immune response, as well as for cancer progression. During mesenchymal cell migration, the Rac1-WAVE-Arp2/3 signalling pathway induces branched actin polymerisation, which protrudes the membrane and allows migration. Fine-tuning the activity of the Rac1-WAVE-Arp2/3 pathway modulates protrusion lifetime and migration persistence. Recently, NHSL1, a novel interactor of the Scar/WAVE complex has been identified as a negative regulator of cell migration in vitro. We here analysed its function in vivo, during zebrafish gastrulation, when *nhsl1b* is expressed in migrating mesodermal cells. Loss and gain of function experiments revealed that *nhsl1b* is required for the proper migration of the mesoderm, controlling cell speed and migration persistence. Nhsl1b localises to the tip of actin-rich protrusions where it controls protrusion dynamics, its loss of function reducing the length and lifetime of protrusions, whereas overexpression has the opposite effect. Within the protrusion, Nhsl1b knockdown increases F-actin assembly rate and retrograde flow. These results identify Nhsl1b as a cell type specific regulator of cell migration and highlight the importance of analysing the function of regulators of actin dynamics in physiological contexts.

Cell migration is a critical phenomenon in both physiological and pathological processes, such as immune cell migration[1], wound healing, or tumour progression, where cancer cells invade the surrounding stroma and initiate metastasis[2]. Cell migration is particularly important during embryogenesis, when cells migrate to build the different tissues and organs of the body[3]. Gastrulation is the earliest developmental stage when cells undergo extensive migrations, to organise into the three germ layers (ectoderm, mesoderm and endoderm), and lay the foundations for the future body plan[4]. In zebrafish, the onset of gastrulation is marked by the internalisation of endodermal and mesodermal precursors, at the margin of the embryo[5–7]. Once inside the embryo, between the yolk syncytial layer and the overlying ectoderm, endodermal cells spread by a random walk[8], while mesodermal cells migrate toward the animal pole[9]. By mid-gastrulation, all these cells engage in convergence-extension movements that will bring them together in the emerging embryonic axis[10]. Gastrulation therefore involves a number of

different cell migrations, and for gastrulation to proceed correctly, it is essential that migratory properties are fine-tuned, so that each cell type undergoes its specific set of movements[11,12]. Quite surprisingly though, very few cell type-specific modulators of cell migration have been reported so far[13].

Most cells migrate by protruding the plasma membrane forward through actin polymerisation. In vitro, on 2D substrates, these protrusions are flat and termed lamellipodia[14], while in vivo, in more complex 3D environments, they take on more complex shapes, and are referred to as actin-rich protrusions[15,16]. The formation of a protrusion requires activation of the small GTPase Rac1, which in turn activates the WAVE complex, which is the main activator of the Arp2/3 complex at the leading edge[17,18]. The Arp2/3 complex generates branched actin networks, which provide the pushing force to protrude the membrane[17]. Modulating this Rac1-WAVE-Arp2/3 pathway directly affects the length and lifetime of protrusions, thereby determining the speed and persistence of migration[19]. Recently,

[1]Laboratoire d'Optique et Biosciences (LOB), CNRS, INSERM, Ecole Polytechnique, Institut Polytechnique de Paris, 91120 Palaiseau, France. [2]Paris-Saclay Institute of Neuroscience, CNRS and University Paris-Saclay, 91400 Saclay, France. ✉e-mail: sophie.escot@polytechnique.edu; nicolas.david@polytechnique.edu

NHSL1 has been identified as a novel regulator of protrusion dynamics and cell migration in vitro[20,21]. NHSL1 is a member of the Nance-Horan syndrome (NHS) family, which in mammals also includes NHS and NHSL2[22,23]. Nance-Horan syndrome is characterised by dental abnormalities, congenital cataracts, dysmorphic features and mental retardation[24]. NHS proteins contain a functional Scar/WAVE homology domain (WHD) at their N-terminus and NHS has been reported to act on actin remodelling and cell morphology in vitro[23]. More recently, two studies identified NHSL1 as a direct binding partner of the WAVE complex, one study suggesting that NHSL1 interacts with the full WAVE complex[20], while the other proposes that NHSL1 may also replace the WAVE subunit within the complex, forming a so-called WAVE Shell Complex[21]. Consistently, NHSL1 colocalises with WAVE complex subunits at the very edge of lamellipodia[20], where both studies proposed that it negatively regulates cell migration, since its knockdown induces an increase in migration persistence of randomly moving cells. However, its precise function may be more complex, as overexpression induced similar defects as the knockdown[20], and NHSL1 appears required to mediate the effect of PPP2R1A, another new interactor of the WAVE Shell Complex, which positively regulates migration persistence[21]. Its effect on the persistence of cells migrating in more physiological contexts has not yet been tested.

In zebrafish, two orthologs of NHS (nhsa and nhsb) and two orthologs of NHSL1 (nhsl1a and nhsl1b) have been identified[25]. nhsl1b has been reported to encode a WHD and is required for the migration of facial branchiomotor neurons[25]. nhsl1b has also been identified during zebrafish gastrulation as a target of Nodal signalling, which is the major inducer of mesendodermal fates, and has accordingly been reported to be expressed at the embryonic margin at the onset of gastrulation[26,27].

We therefore sought to analyse the in vivo function of nhsl1b by characterising its role during zebrafish gastrulation. We show that nhsl1b is expressed in ventral, lateral and paraxial mesodermal cells, and is required for their proper migration toward the animal pole, controlling migration speed and persistence. Nhsl1b localises at cell-cell contacts and at the very tip of actin-rich protrusions where it positively regulates their length and lifetime. In the protrusion, Nhls1b knockdown increases F-actin assembly rate and retrograde flow.

## Results

### nhsl1b is expressed in mesodermal cells during gastrulation

Using in situ hybridisation, we analysed the expression pattern of nhsl1b in early zebrafish embryos. Maternal expression was visible at the 1-cell stage (Fig. 1A, first column). Consistent with previous reports[26], at the onset of gastrulation, nhsl1b appeared to be specifically expressed at the margin of the embryo (Fig. 1A, second column), in the region encompassing mesendodermal precursors[28]. By mid-gastrulation, nhsl1b expression was detected in the involuted ventral, lateral and paraxial mesendoderm, as well as in the most posterior chorda mesoderm (Fig. 1A, third column). During somitogenesis, nhsl1b expression was restricted to somites (Fig. 1A, fourth column).

It has been previously reported that nhsl1b expression during gastrulation is under the control of Nodal signalling, as it is lost in MZoep embryos, which are lacking Nodal signalling, and is induced by global overexpression of ndr1 (Nodal related 1)[26]. We confirmed these results using cell transplants, to avoid overexpressing Nodal signals in the entire embryo. A few Ndr1 expressing cells were transplanted to the animal pole of a wildtype host embryo. Consistent with previous reports, we observed nhsl1b expression among transplanted cells, confirming that its expression is inducible by Nodal (Fig. 1B).

Nodal signals are responsible for induction of both endodermal and mesodermal cells. Endodermal cells being few in number, the expression profile of nhsl1b is compatible with an expression in both endo and mesodermal cells, or specifically in mesodermal cells. To discriminate, we directly tested for expression in endodermal cells. Donor embryos were injected with sox32 mRNA which directs cell towards the endoderm lineage[29]. Induced endodermal cells were transplanted to the animal pole of a wildtype host embryo (Fig. 1C). The endodermal identity of the transplanted cells was confirmed by expression of the endodermal marker sox17 (Fig. 1C, bottom).

We did not observe any nhsl1b expression in the transplanted cells, suggesting that nhsl1b is not expressed in endodermal cells and specifically expressed in mesodermal ones. Consistently, we analysed available scRNAseq data[30], and observed that, at the onset of gastrulation, the gene with the most correlated expression profile to nhsl1b is tbxta (r = 0.21), a mesodermal marker, while there was no correlation to endodermal markers (sox32, r = −0.01; sox17, r = −0.02). Finally, we carefully analysed the dorsal marginal expression, which could overlap with the region containing forerunner cells, precursors of Kupffer's vesicle, which are not mesodermal[31]. We performed double in situ hybridisation for nhsl1b and sox17 at mid gastrulation (75% epiboly) and bud stage (Fig. 1D) and observed that sox17-positive dorsal forerunner cells do not express nhsl1b. These results suggest that nshl1b expression during gastrulation is exclusive to mesodermal cells.

### nhsl1b regulates mesodermal migration

To investigate the function of nhsl1b during lateral mesodermal cell migration, we used the zebrafish transgenic line Tg(tbx16:EGFP) which labels mesodermal cells. We knocked-down nhsl1b expression using a previously published antisense morpholino directed against the start codon (MO-1)[25]. We observed that upon nhsl1b knockdown, tbx16 expressing mesodermal cells internalise at the margin of the embryo and initiate their animalward movement (Fig. 2A, t = 0, Supplementary Movie 1). However, their migration toward the animal pole appeared to be slower than in embryos injected with a control morpholino. To quantify this, we selected control and nhsl1b morphant embryos at early gastrulation (60% epiboly), in which the front of the lateral mesoderm was 150 µm ±50 µm away from the margin, and quantified the progression of the front over the following hour. In control embryos, the margin to mesoderm front distance increased by 138 µm ±35 µm. In nhsl1b morphants it increased by only 95 µm ±65 µm (Fig. 2A, B, Supplementary Movie 1). Since the measured margin-to-front distance depends not only on mesodermal progression but also on margin progression, we ensured that the observed defect was not reflecting a defect in epiboly. First, we analysed the progression of epiboly in control and nhsl1b morphants by measuring the distance from the animal pole to the margin, and did not observe a significant difference (Supplementary Fig. S1A, C). Second, we quantified the progression of the mesoderm front, by directly measuring the distance from the mesoderm front to the animal pole (Supplementary Fig. S1B, D). This measure is less accurate than measuring the margin to front distance, as it is biased by the embryo sphericity, but it is independent from the epiboly movement and confirmed that the mesoderm front progresses slower in nhsl1b morphants (Supplementary Fig. S1D).

To ensure the specificity of the observed phenotype, we repeated the experiments with a second, independent, morpholino (MO-2 Nhsl1b) and observed a similar reduction in the progression of the mesoderm front (Fig. 2A, B). Importantly, as this second morpholino is targeting the 5'UTR of nhsl1b, we could perform rescue experiments, co-injecting the morpholino and morpholino-insensitive nhsl1b mRNAs. This restored mesoderm progression to values comparable to control embryos, confirming that the observed phenotype is due to the loss of nhsl1b function and not to off-target effects (Fig. 2B and Supplementary Fig. S2A).

To further confirm the specificity of the observed phenotype, we applied an independent strategy to knock-down nhsl1b, using the CRISPR/Cas13d system. This system has been shown to specifically degrade targeted mRNAs in various organisms and has been successfully applied to zebrafish[32]. Co-injecting cas13d mRNA and a mix of 3 guide RNAs targeting nhsl1b reduced nhsl1b transcripts level by 77% (Supplementary Fig. S3A). Similar to the morpholino knockdown, CRISPR/Cas13d knock-down of nhsl1b reduced the progression of the mesoderm front by an average of 46 µm compared to control embryos, injected with cas13d mRNAs alone (Supplementary Fig. S3B, C). Taken together, these results indicate that nhsl1b is required for proper mesodermal migration toward the animal pole during gastrulation.

### Nhsl1b regulates cell migration persistence and speed

In vitro, nhsl1 has been shown to control cell migration speed and persistence, its knock-down increasing both[20]. To understand how nhsl1b controls

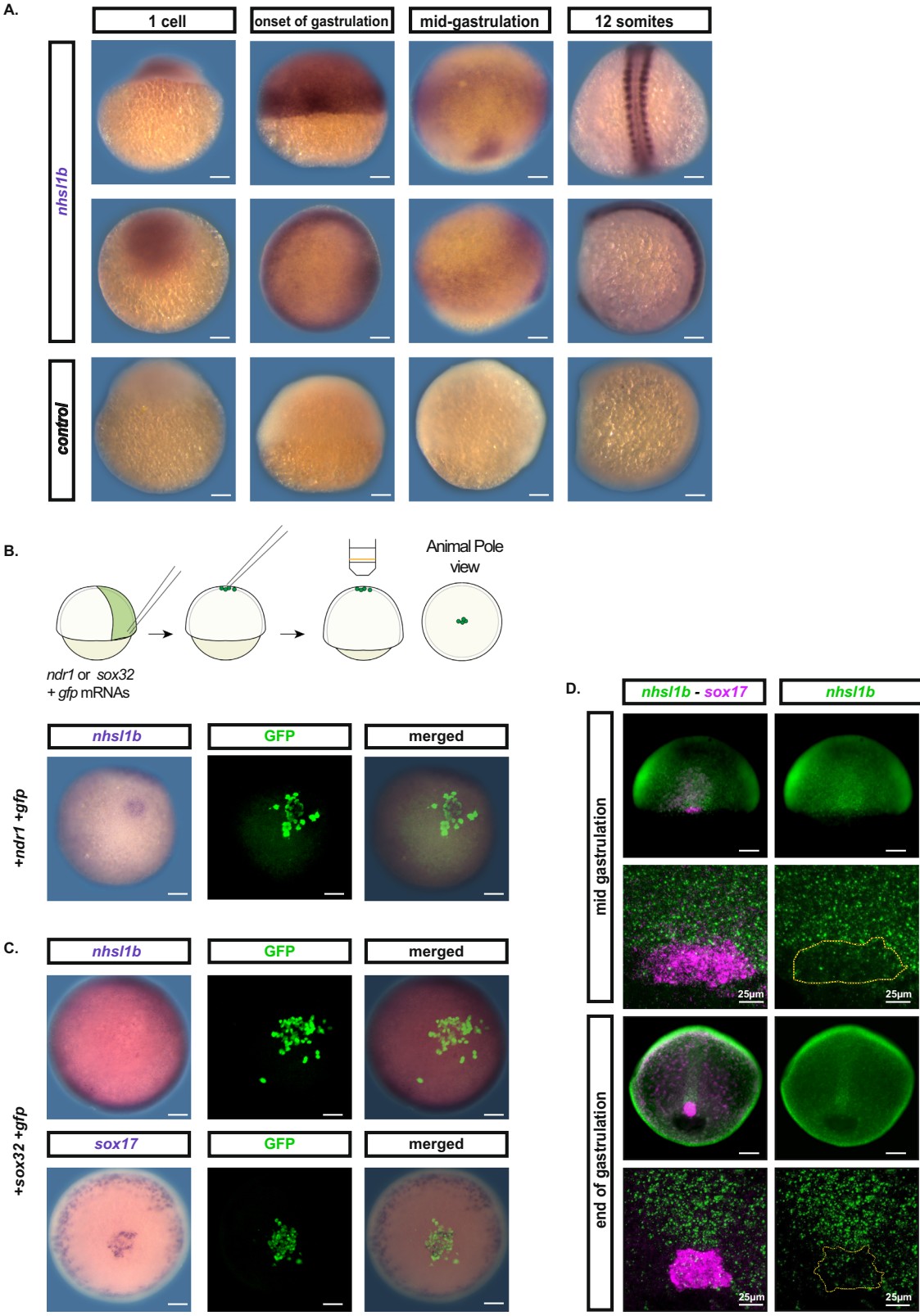

**Fig. 1 | *nhsl1b* is expressed in mesodermal cells during gastrulation. A** Whole-mount in situ hybridisation with *nhsl1b* or control probes, on embryos fixed at the 1-cell, shield (onset of gastrulation), 75% epiboly (mid-gastrulation) and 12-somite stages. **B**, **C** Whole mount in situ hybridisation with *nhsl1b* and *sox17* probes and subsequent GFP immunostaining. **B** *nhsl1b* is expressed in induced mesendodermal cells. *N* = 4 experiments. **C** No expression of *nhsl1b* is observed in induced endo-dermal cells. *N* = 3 experiments. **D** Whole-mount double in situ hybridisation with *sox17* (red) and *nhsl1b* (green) probes at 75% epiboly (end of gastrulation) and bud stages. *nhsl1b* is not expressed in the *sox17* expressing fore-runner cells. Scale bar 100 µm unless specified.

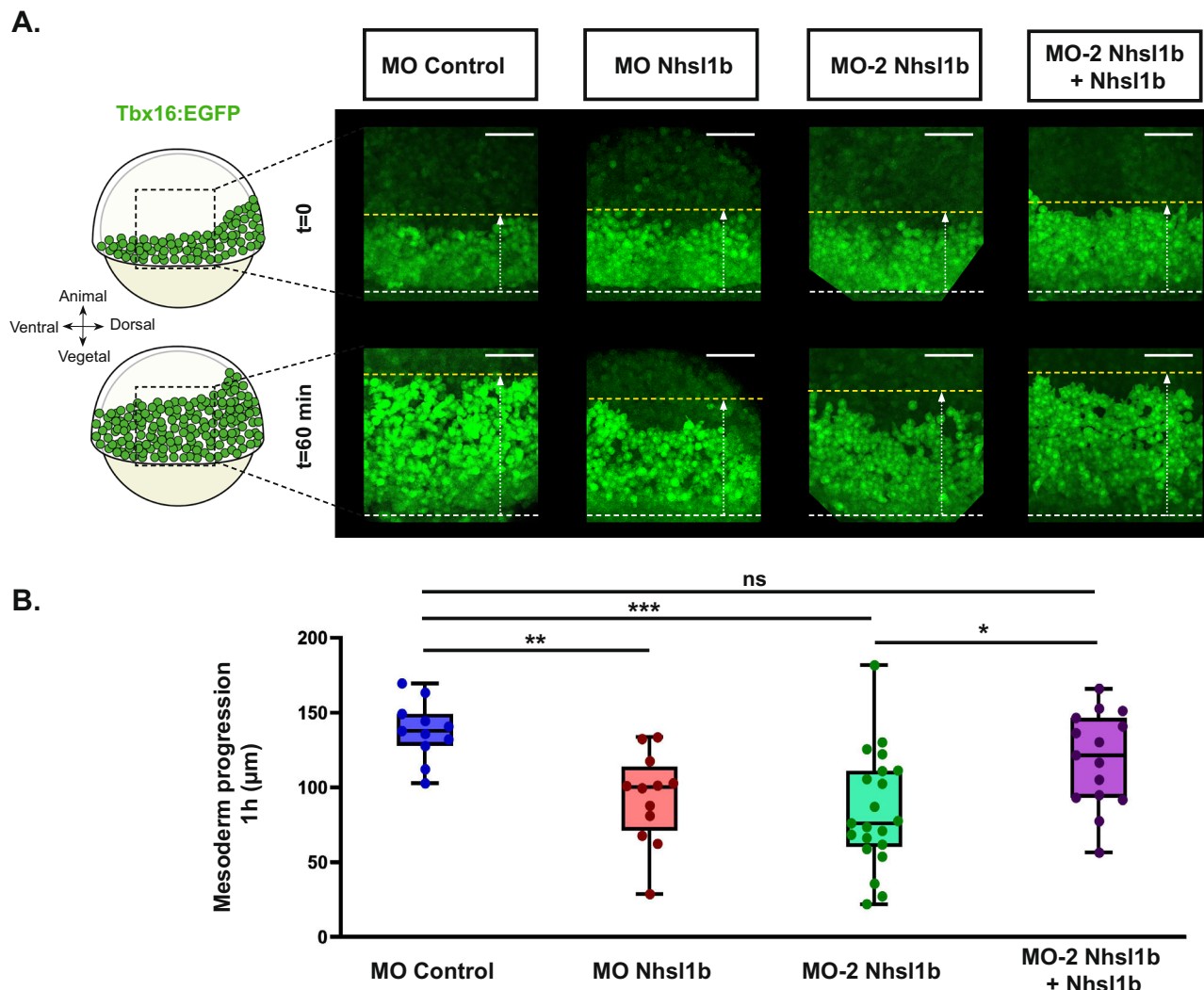

**Fig. 2 | nhsl1b knockdown affects lateral mesoderm migration. A** Representative lateral views of *Tg(tbx16:EGFP)* embryos, at early gastrulation (60% epiboly; t = 0) and 1 hour later, in different conditions. White dashed lines indicate the position of the margin of the blastoderm, yellow dashed lines indicate the position of the front of the migrating mesoderm. Mesoderm progression was measured as the variation in distance between these two lines (see also Supplementary Fig. S1). Scale bar 100 μm. **B** Quantification of the lateral mesoderm progression in MO Control injected embryos (*n* = 11 embryos), MO Nhsl1b injected embryos (*n* = 12 embryos), MO-2 Nhsl1b injected embryos (*n* = 21 embryos), MO-2 Nhsl1b and *nhsl1b* mRNA co-injected embryos (*n* = 15 embryos). *N* = 16 experiments in total with more than 4 experiments per conditions. Kruskal–Wallis test followed by Dunn's test. Adjusted *p*-values: MO Control vs MO Nhsl1b: 0.0093 **; MO Control vs MO-2 Nhsl1b 0.0002 ***; MO-2 Nhsl1b vs MO-2 Nhsl1b + Nhsl1b 0.0218 *; MO Control vs MO-2 Nhsl1b + Nhsl1b 0.6481 ns.

progression of the mesodermal layer, we analysed the effect of its knock-down on the movement of individual mesodermal cells. All nuclei were labelled with H2B-mCherry in *Tg(tbx16:EGFP)* embryos, and lateral mesodermal cells were tracked for 1 hour starting from early gastrulation (60% epiboly), in control and *nhsl1b* morphants (Fig. 3A, Supplementary Movie 2). *nhsl1b* knock-down induced a moderate but significant reduction of the instantaneous cell speed (Fig. 3B). We measured migration persistence as the directionality ratio (ratio of the straight-line distance to the total trajectory path length of each track, over a defined time interval of 6 min), and observed a reduction of persistence in *nhsl1b* knocked-down cells (Fig. 3C, E, F). This was confirmed by analysing the directional autocorrelation[33], a measure of how the angles of displacement vectors correlate with themselves upon successive time points (Fig. 3D). Plotting cell tracks colour-coded for persistence revealed that in control embryos, animal most cells are more persistent than more posterior ones, possibly because they are free to move animalward, without bumping into neighbours. In *nhsl1b* morphants, the reduction in persistence is particularly visible in these cells (Fig. 3E, F). These results show that knocking down *nhsl1b* affects the speed and persistence of migration, both effects accounting for the observed reduced animalward movement of the mesodermal layer.

## The level of Nhsl1b needs to be fine-tuned for optimal migration

To further explore the role of *nhsl1b* in mesodermal cell migration we overexpressed *nhsl1b* by injecting mRNA in *Tg(tbx16:EGFP)* embryos and analysed lateral mesodermal cell movements. As a control, embryos were injected with an equal amount of *lacZ* mRNA. Mesodermal cells from *nhsl1b* mRNA injected embryos internalised at the margin and migrated toward the animal pole (Fig. 4A, t = 0, Supplementary Movie 1). However, as observed in the *nhsl1b* knockdown, they migrated slower: over one hour, the distance between the embryonic margin and the mesoderm front increased by 79 μm ±36 μm in *nhsl1b* overexpressing embryos, compared to 120 μm ±51 μm in controls (Fig. 4A, B, Supplementary Movie 1). To investigate the cause of this delay, we labelled nuclei with H2B-mCherry in *Tg(tbx16:EGFP)* embryos and tracked lateral mesodermal cells. Similar to *nhsl1b* knock-down, the overexpression of *nhsl1b* induced a reduction of cell speed (Fig. 4C) compared to controls. It also induced a reduction of cell persistence, measured as the directionality ratio or as the directional auto-correlation (Fig. 4D, E). Overexpression of *nhsl1b* thus results in migration defects very similar to those induced by its knockdown, suggesting that *nhsl1b* expression level needs to be tightly controlled to achieve optimal cell

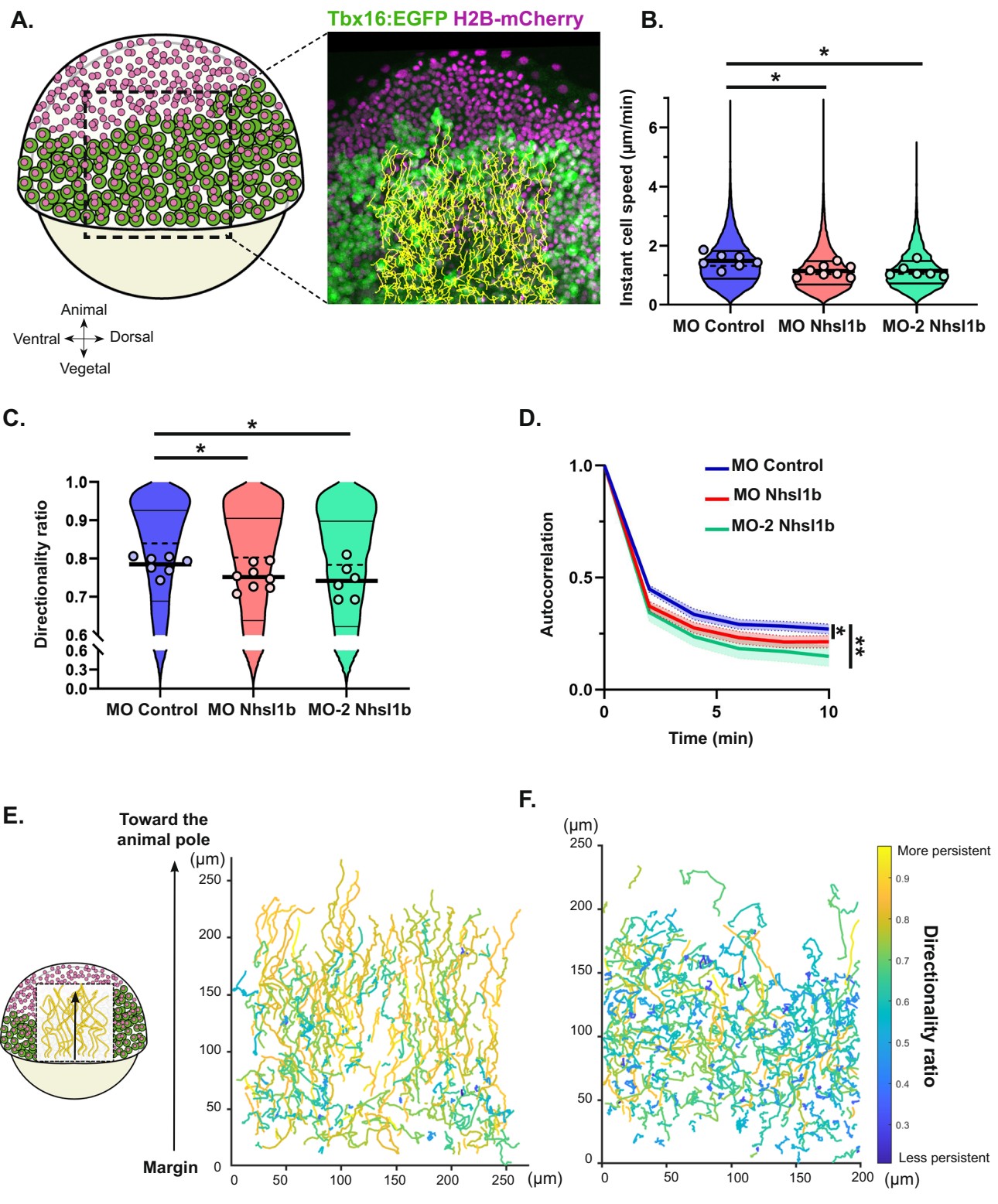

**Fig. 3 | *nhsl1b* knockdown reduces cell speed and persistence. A** Representative image of mesodermal cell nuclei tracking, in a *Tg(tbx16:EGFP)* embryo expressing H2B-mCherry. **B**, **C** Instant cell speed and directionality ratio. Circles on the violin plot indicate mean per embryo. Likelihood ratio test of a linear mixed effects model with treatment as a fixed effect and embryos as a random effect against a model without the fixed effect. Adjusted *p*-values: (**B**) MO Control vs MO Nhsl1b: 0.0100 *; MO Control vs MO-2 Nhsl1b: 0.0131 *; (**C**) MO Control vs MO Nhsl1b: 0.0301 *; MO Control vs MO-2 Nhsl1b: 0.0301 *. **D** Directional autocorrelation. Likelihood ratio test of a non-linear mixed effects model (see methods) with treatment as a fixed effect and embryos as a random effect against a model without the fixed effect. Adjusted *p*-values: MO Control vs MO Nhsl1b: 0.045 *; MO Control vs MO-2 Nhsl1b: 0.0054 *. MO control (*n* = 5), MO Nhsl1b (*n* = 8) and MO-2 Nhsl1b (*n* = 6) injected embryos. **E**, **F** Representative cell trajectories of mesodermal cells in MO control and MO Nhsl1b injected embryos, colour-coded for cell migration persistence (directionality ratio). x-axis represents distance along the dorsal-ventral axis, y-axis represents distance along the animal-vegetal axis.

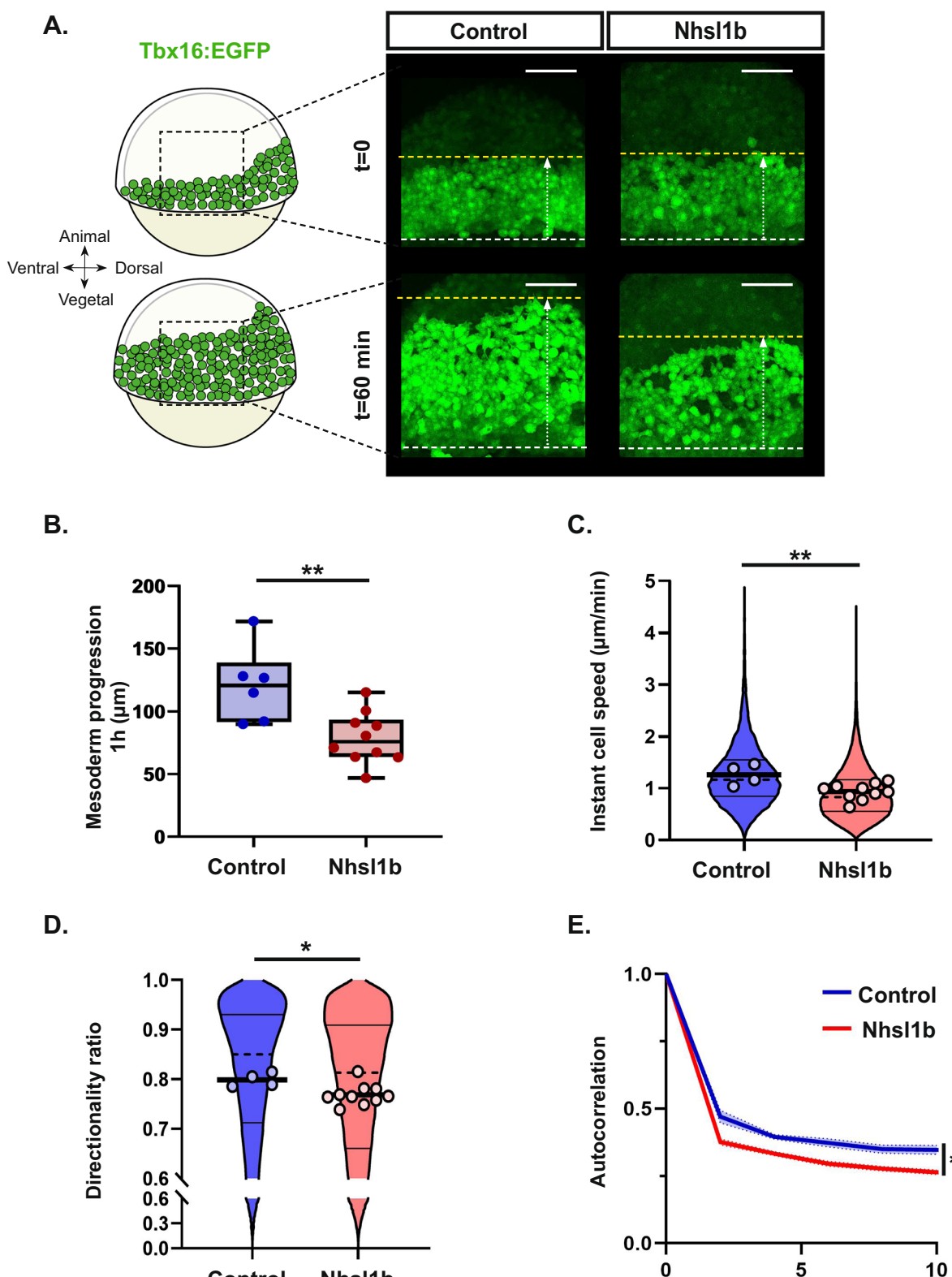

**Fig. 4 | *nhsl1b* overexpression affects lateral mesodermal migration, reducing cell speed and persistence. A** Representative lateral views of *Tg(tbx16:EGFP)* embryos, at early gastrulation (60% epiboly; t = 0) and 1 hour later, in embryos injected with *lacZ* (control, *n* = 6) or *nhsl1b (n* = 10) mRNAs. Dashed lines indicate the positions of the front of the migrating mesoderm on the first (yellow) and last (white) frames. Mesoderm progression was measured as the distance between these two lines (see also Supplementary Fig. S1). Scale bar 100 μm. **B** Quantification of the lateral mesoderm progression in control and *nhsl1b* mRNA injected embryos. *N* = 6

experiments. Mann–Whitney. *p*-value: Control vs Nhsl1b: 0.0075*, **C, D** Instant cell speed and directionality ratio. Circles on the violon plot indicate mean per embryo. Likelihood ratio test of a linear mixed effects model with treatment as a fixed effect and embryos as a random effect against a model without the fixed effect. *p*-values: 0.0029 ** and 0.0111 *. **E** Directional autocorrelation. Likelihood ratio test of a non-linear mixed effects model (see methods) with treatment as a fixed effect and embryos as a random effect against a model without the fixed effect. *p*-value: 0.03589 *. Control (*n* = 4) and *nhsl1b* (*n* = 10) mRNA injected embryos.

migration. Accordingly, overexpression phenotypes and rescue experiments were very sensitive to *nhsl1b* mRNA doses (Supplementary Fig. S2).

## Nhsl1b localises to the tip of actin-rich protrusions

Knockdown and overexpression experiments revealed a role for *nhsl1b* in controlling migration speed and persistence. For cells migrating in vitro, on a 2D substrate, both speed and persistence are largely determined by the dynamics of the lamellipodium[19], and NHSL1 has been shown to localise to the very edge of the lamellipodium in cultured cells[20,21].

To study the localisation of Nhsl1b in migrating mesodermal cells, we generated a mNeongreen tagged version of the zebrafish Nhsl1b protein. We first analysed its subcellular localisation in mesodermal cells plated

in vitro, as precise subcellular observations are easier in these conditions than in the intact embryo. *Nhls1b-mNeongreen* and *Lifeact-mCherry* (a marker for F-actin) mRNAs were injected at the 4-cell stage. At early gastrulation (60% epiboly stage), some mesodermal cells were dissected and plated on a glass bottom chamber. Some of the cells in direct contact with the glass extend large flat extensions, resembling lamellipodia. Nhsl1b localised all along the edge of these lamellipodia, as observed in melanoma cell line[20,21] (Fig. 5A, Supplementary Movie 3) and remained at the edge while the protrusion extended (see kymograph in Fig. 5A).

In vivo, in complex 3D environments, migrating cells such as the lateral mesoderm do not form characteristic lamellipodia, but rely on functionally equivalent actin-rich protrusions[16]. Under our in vitro conditions of culture,

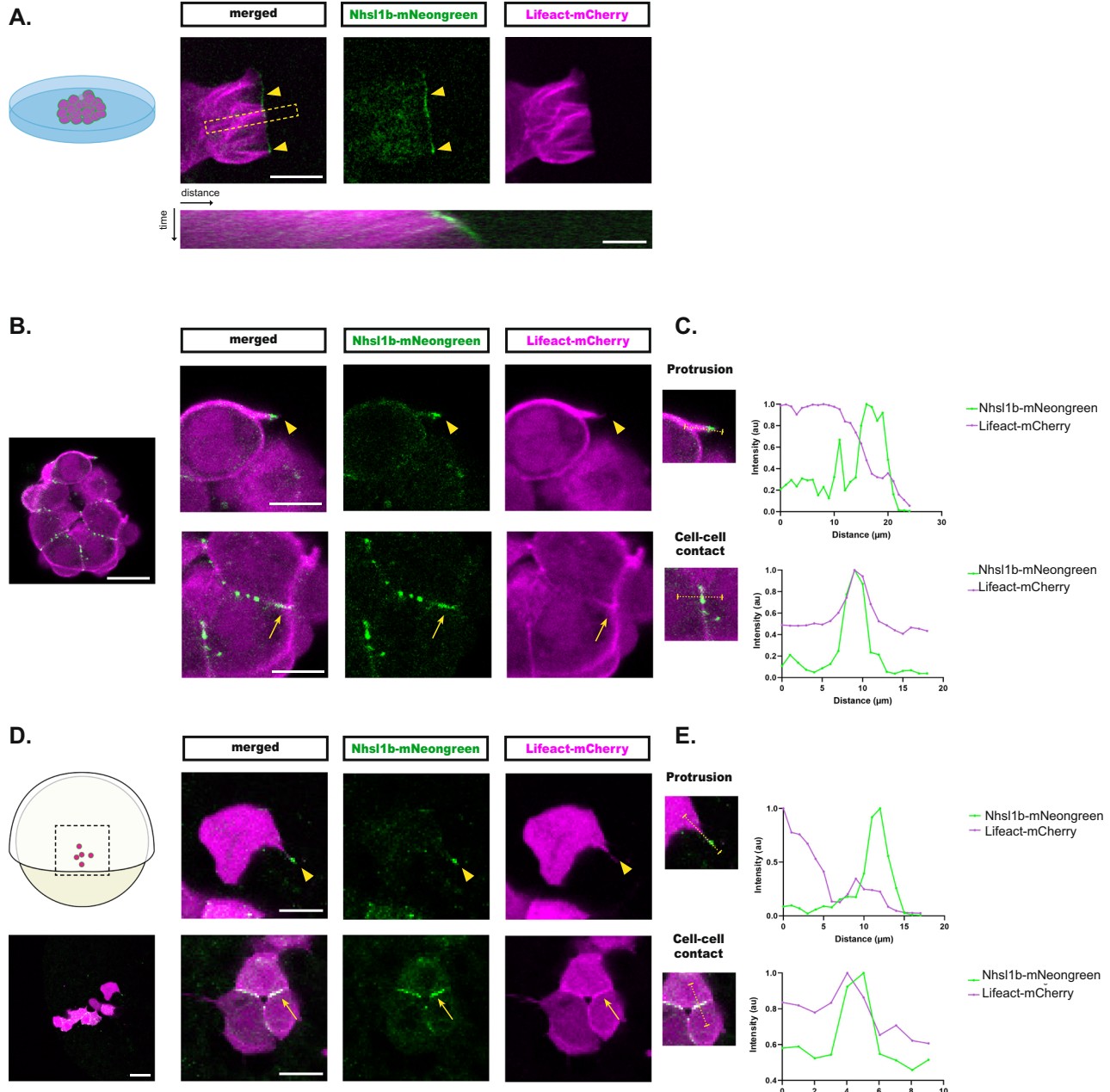

**Fig. 5 | Nhsl1b localises at cell-cell contacts and at the tip of actin-rich protrusions.** Nhsl1b-mNeongreen and Lifeact-mCherry expressing mesodermal cells plated on a coverslip (**A–C**) or in vivo, transplanted in a non-labelled host embryo (**D, E**). **A** Nhsl1b localises at the very edge of the lamellipodium (arrowhead), where it stands during lamellipodium progression, as revealed by the kymograph (performed over the region delineated by the dashed yellow box). **B, C** In cultured cell clusters, Nhsl1b localises at cell-cell contacts (arrow) and at the very tip of actin-rich protrusions (arrowhead). **D, E** Similar localisations are observed in mesodermal cells in vivo. **C, E** Quantification of the max-normalised intensity of Nhsl1b-mNeongreen and Lifeact-mCherry along the segments indicated in the inset. Scale bars: (**A**) 10 μm; 2 μm, (**B**) 20 μm; 10 μm, (**D**) 40 μm; 20 μm.

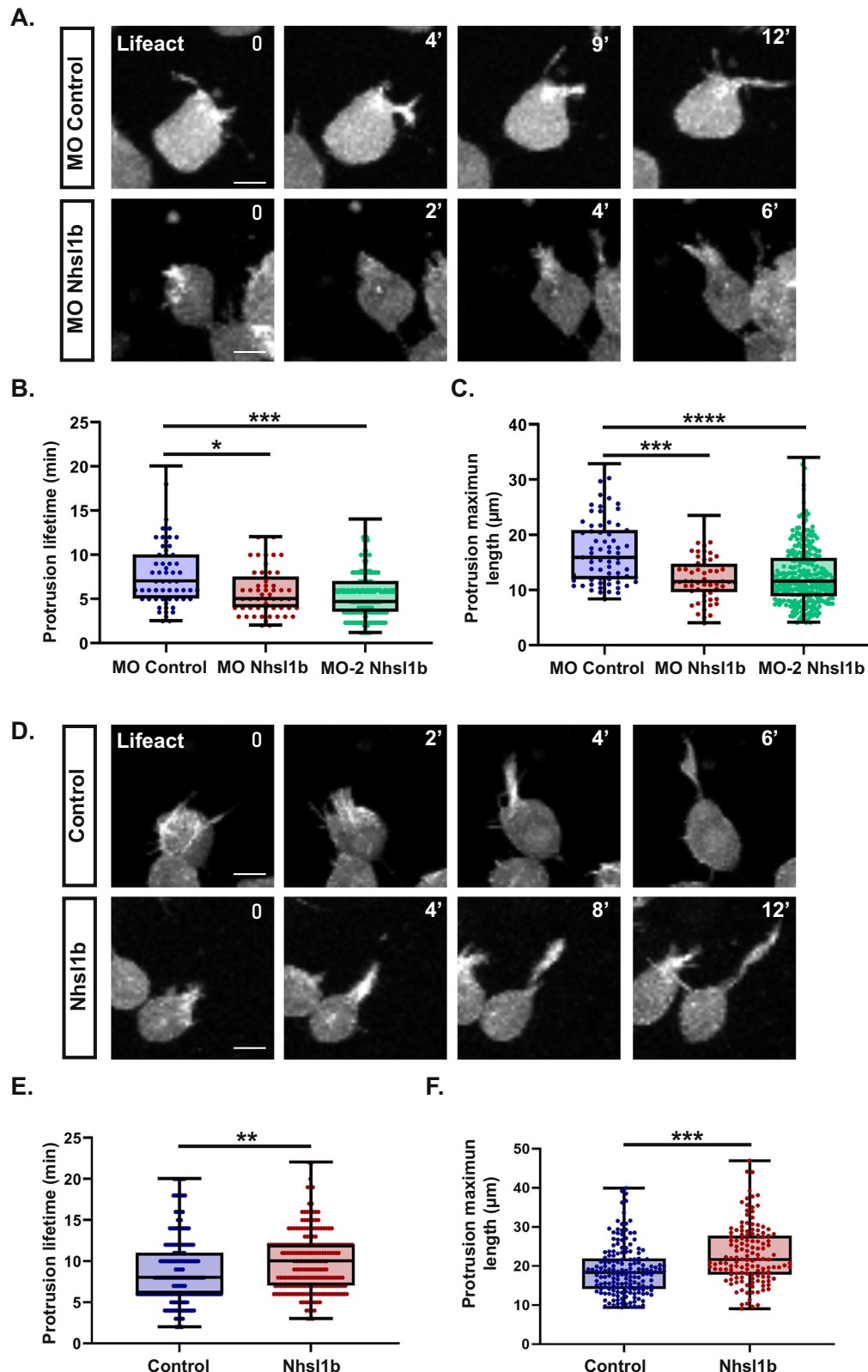

some cells form clusters but continue to produce actin-rich protrusions, as they do in the embryo. We analysed the localisation of Nhsl1b-mNeongreen in these clusters of cells. We observed distinct Nhsl1b-mNeongreen dots along cell-cell contacts that often, but not always, co-localised with Lifeact. Focusing on actin-rich protrusions, we observed Nhsl1b-mNeongreen accumulations at the very tip of the protrusions (Fig. 5B, C).

To confirm these observations in vivo, we used mosaic embryos, to label only a few cells, which is key for proper observation of cell protrusions. *Nhsl1b-mNeongreen* and *Lifeact-mCherry* mRNAs were injected into donor embryos at the 4-cell stage, and, at the sphere stage, a few marginal cells were transplanted to the margin of unlabelled host embryos. At early gastrulation (60% epiboly), we picked embryos with labelled cells in the lateral mesoderm

**Fig. 6 | Nhsl1b regulates protrusion dynamics. A** Protrusions of mesodermal cells injected with *Lifeact-mCherry* mRNAs and with a MO control or a MO Nhsl1b, transplanted in the mesoderm of a non-labelled embryo. Selected time points showing the protrusion elongation. **B**, **C** Quantification of the lifetime and maximum length of protrusions. Likelihood ratio test of a linear mixed effects model with treatment as a fixed effect and cells as a random effect against a model without the fixed effect. Adjusted *p*-values: (**B**) MO Control vs MO Nhsl1b: 0.0309*; MO Control vs MO-2 Nhsl1b: 0.0011 ***; (**C**) MO Control vs MO Nhsl1b: 0.0001***; MO Control vs MO-2 Nhsl1b: 4,89E−06****. MO Control (*n* = 5 embryos; *n* = 12 cells), MO Nhsl1b (*n* = 6 embryos; *n* = 14 cells) and MO-2 Nhsl1b (*n* = 7 embryos, 49 cells). **D** Protrusions of mesodermal cells injected with *Lifeact-mCherry* mRNAs and with *lacZ* (control) or *Nhsl1b* mRNAs, transplanted in the mesoderm of a non-labelled embryo. Selected time points showing the protrusion elongation. **E, F** Quantification of the lifetime and maximum length of protrusions. Likelihood ratio test of a linear mixed effects model with treatment as a fixed effect and cells as a random effect against a model without the fixed effect. *p*-values: **E** Control vs Nhsl1b: 0.0094**; (**F**) Control vs Nhsl1b: 0.00012***. Control (*n* = 5 embryos, 30 cells) or Nhsl1b (*n* = 6 embryos, 28 cells). Scale bars 20 μm.

and analysed Nhsl1b localisation. Similar to what we observed in plated cells, Nhsl1b-mNeongreen localised at cell-cell contacts and at the very tip of actin-rich protrusions (Fig. 5D, E).

## Nhsl1b increases the stability of protrusions

Given its localisation at the tip of actin-rich protrusions, and as it has been shown to regulate actin dynamics in vitro, we wondered whether Nhsl1b could regulate cell migration by modulating protrusion dynamics. To address this, we quantified the lifetime and maximum length of protrusions in Lifeact-mCherry expressing mesodermal cells, transplanted into wildtype embryos (Fig. 6A, D). Strikingly, *nhsl1b* knockdown induced a reduction in average protrusion duration (2 min i.e. 25% reduction for MO Nhsl1b; 2.5 min i.e. 33% reduction for MO-2 Nhsl1b) compared to MO control injected cells (Fig. 6B), as well as a reduction in average maximal length (5 μm i.e. 30% reduction for MO Nhsl1b; 4 μm i.e. 25% reduction for MO-2 Nhsl1b) (Fig. 6C). Conversely, in cells overexpressing *nhsl1b*, protrusion lifetime was increased by an average of 1 min (Fig. 6E), and protrusions were on average 4 μm longer than in control cells (Fig. 6F). Altogether, these results indicate that Nhsl1b regulates the stability of protrusions during mesodermal cell migration and that protrusion dynamics must be fine-tuned for efficient migration.

## Nhsl1b regulates actin assembly and retrograde flow in the protrusions

To gain a better understanding of how Nhsl1b affects protrusion dynamics, we quantified actin dynamics parameters during protrusion extension. A few MO Control or MO Nhsl1b Lifeact-mNeongreen expressing mesodermal cells were transplanted into wildtype hosts and imaged at high temporal resolution (Fig. 7A, Supplementary Movie 4). Kymographs of the extending protrusion revealed no significant difference in the speed of forward protrusion of the membrane (Fig. 7B, C). Actin retrograde flow, however, was significantly increased by *nhsl1b* knockdown (Fig. 7D). Consistently, the actin polymerisation rate, which can be calculated as the magnitude of the difference between the protrusion speed vector and the flow speed vector[34], was increased upon *nhsl1b* knockdown (Fig. 7E). These results show that Nhsl1b regulates actin assembly rate and retrograde flow speed in protrusions of mesodermal cell.

## Discussion

Very recently, two studies have identified NHSL1 as a regulator of cell migration in vitro, in the mouse melanoma cell line B16-F1[20] and in non-malignant human breast epithelial cells[21]. Here, we report the characterisation of its in vivo function, in migrating mesodermal cells during zebrafish gastrulation. We reveal that precisely controlled levels of *nhsl1b* expression are required to ensure proper mesoderm migration, as both *nhsl1b* knockdown and overexpression result in a reduction in cell speed and migration persistence. Nhsl1b localises to the very tip of actin-rich protrusions, and controls their stability, Nhsl1b promoting length and duration of protrusions. Within the protrusion, Nhsl1b appears to limit F-actin assembly and F-actin retrograde flow speed.

A previous report by Walsh et al. has described a specific function of Nhsl1b in the migration of facial branchiomotor neurons in the zebrafish embryo[25]. This study focused on the *fh131* mutant, a mutant identified in a forward genetic screen looking for genes affecting facial branchiomotor neurons. The *fh131* mutant harbours a point mutation on exon 6 creating a

premature stop codon, likely leading to the production of a C-terminally truncated protein, still bearing the Wave Homology Domain (exon 1). In maternal zygotic *fh131* mutant embryos, the authors did not report any defect before the neuronal migration defects. The molecular nature of the *fh131* allele, and the absence of a gastrulation phenotype, suggest that this allele may not be a null, which is why we chose to study the role of Nhsl1b through two independent knock-down approaches (morpholinos and CRIPR/Cas13d) affecting the entire Nhsl1b protein.

Our in vivo analysis of Nhls1b function is overall very consistent with in vitro observations. We observed that Nhsl1b localises to cell-cell contacts, and, as seen in vitro[20,21], to the tip of actin-rich protrusions, which are the in vivo functional equivalents of lamellipodia. As in vitro, our results clearly establish a role of Nhsl1b in controlling cell migration speed and persistence, and in modulating actin dynamics. Very interestingly, there are nevertheless some differences between in vitro and in vivo results. In particular, we observed that Nhsl1b knockdown leads to a reduction in cell speed and persistence, and an increase in F-actin assembly and retrograde flow, when it seems to have opposite effects in cultured cells. A number of reasons could account for these differences between in vitro and in vivo situations.

First, regarding F-actin assembly, Law et al. reported that NHSL1 knockdown reduces actin assembly in the lamellipodium[20]. However, they also observed that NHSL1 knockdown increases Arp2/3 activity. This increase in Arp2/3 activity nicely fits with our observation that Nhsl1b knockdown increases F-actin assembly rate. The difference between the two systems may arise, as suggested by Law et al., from differences in actin filament density and membrane tension which influence F-actin assembly rate[34]. Wang et al. reported that NHSL1 knockdown increases migration persistence in vitro. In the meantime, they observed that NHSL1 is required to mediate the effect of PPP2R1A loss of function, which reduces migration persistence[21]. One possible explanation for these contradictory effects is that NHSL1 appears to participate in two distinct WAVE complexes, one that includes WAVE itself[20], and one in which NHSL1 replaces WAVE[21], and may have different functions in these two conditions. The phenotype resulting from NHSL1/Nhsl1b knockdown would then depend on the balance between these two functions, which may not be the same for cells in culture and cells in physiological conditions. Consistent with the idea that NHSL1/Nhsl1b may play opposite roles, we observed, as in in vitro studies, that Nhsl1b knockdown and overexpression lead to similar effects on persistence, demonstrating that both in vitro and in vivo, an optimal level of NHSL1/Nhsl1b is required to fine-tune cell migration.

A second major difference between the in vitro and in vivo situations is that both in vitro studies focused on randomly migrating cultured cells, whereas, in vivo, mesoderm migration is directed toward the animal pole. In directed migration, membrane protrusions at the leading edge must be sustained, for efficient migration, but must also be able to retract, to allow cells to turn and respond to guidance cues. Increasing cell persistence can therefore limit the ability of cells to respond to guidance cues, and result in an effective reduction of migration persistence. We previously reported such an observation, analysing the Arp2/3 inhibitor Arpin. Inhibition of Arpin in randomly moving cells increases their persistence. Inhibition of Arpin in vivo, in dorsal mesodermal cells migrating toward the animal pole, results in a reduced persistence, as cells cannot efficiently fine-tune their orientation in response to guidance cues[35].

These in vitro/in vivo differences highlight the importance of analysing cell migration regulators in their physiological context, as the complex and

**Fig. 7 | *nhsl1b* knockdown increases F-actin retrograde flow and assembly rate. A** Selected time points from a high temporal resolution time-lapse showing growing protrusions in MO Control and MO Nhsl1b injected mesodermal cells expressing Lifeact-mNeongreen. Dashed yellow box indicate the regions used to generate the kymographs in (**B**). **B** Kymographs colour-coded to highlight the forward and backward moving pixels. The yellow dashed line corresponds to the extending front of the protrusion, the white dashed line corresponds to the actin retrograde flow. Quantification of the protrusion extension speed (**C**), retrograde flow speed (**D**), and F-actin assembly rate (**E**). Likelihood ratio test of a linear mixed effects model with treatment as a fixed effect and cells as a random effect against a model without the fixed effect. *p*-values: (**C**) MO Control vs MO Nhsl1b: 0,8362 ns.; (**D**) MO Control vs MO Nhsl1b: 0.0051**; (**E**) MO Control vs MO Nhsl1b: 0.0248*. MO Control (*n* = 84 cells in 14 embryos) and MO Nhsl1b (*n* = 89 cells in 17 embryos). Scale bars 2 μm.

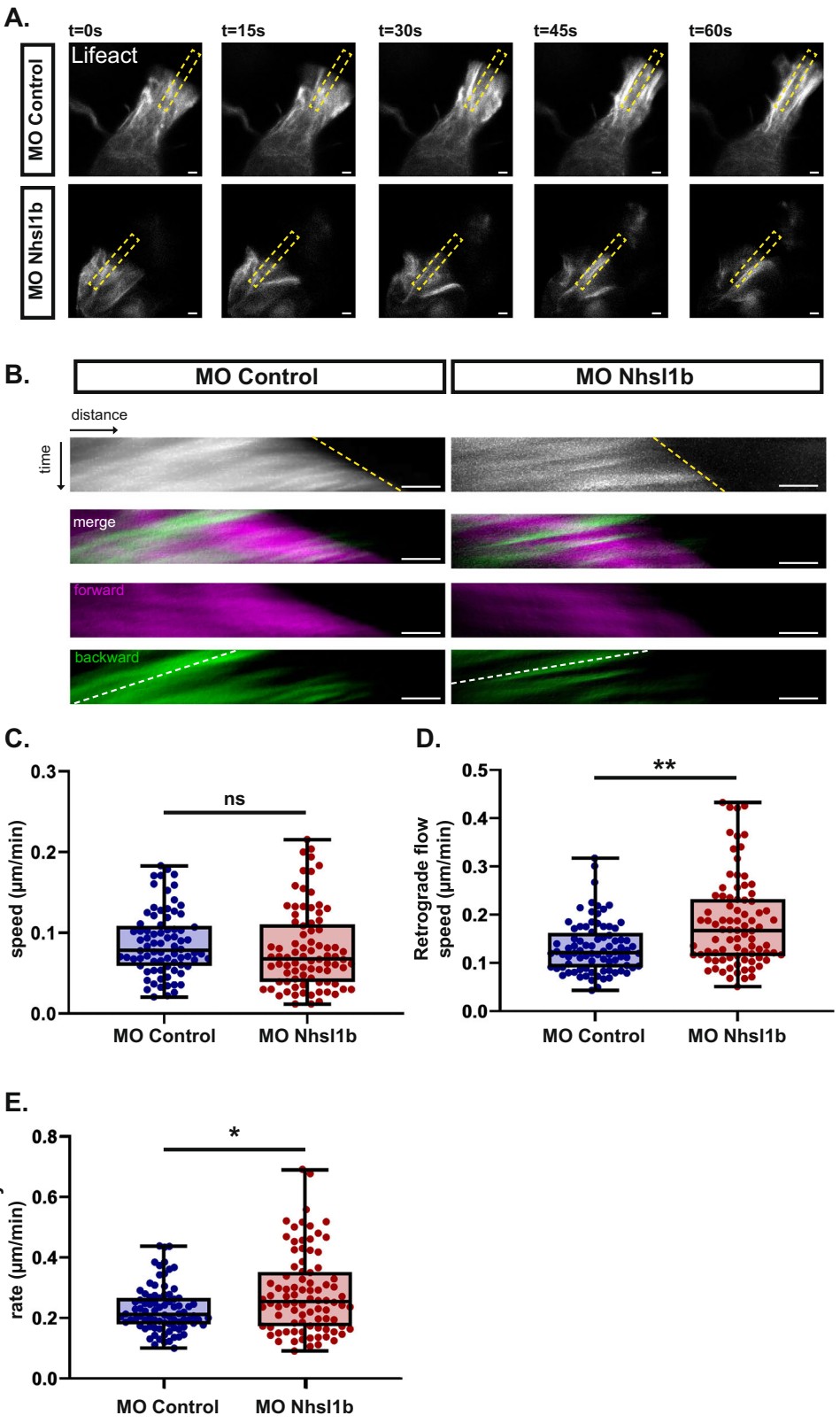

partially understood feedbacks that regulate actin assembly and migration persistence may lead to different outcomes depending on the cell state and environment. We observed that in Nhsl1b knockdown, which leads to a reduction in protrusion length and in cell persistence there is an increase of F-actin retrograde flow. While these observations are consistent with some previous in vitro studies that have observed an increase of F-actin retrograde flow in non-persistently protruding cells[36,37] they also contrast with the proposed law that increased retrograde flow stabilises cell polarity and increases persistence[38]. This suggests that the relationship between retrograde flow and migration persistence is more nuanced than previously appreciated and reinforces the need to consider cellular and environmental context in understanding migration regulation.

In line with this idea that complex feedbacks regulate migration persistence, it is very striking that Nhsl1b knockdown, which reduces protrusion length and lifetime, results in a similar migration phenotype as Nhsl1b overexpression, which increases protrusion length and lifetime. This clearly highlights that cell behaviour arises from a precisely set amount of protrusivity, as very recently demonstrated for the internalisation of mesendodermal cells[39]. In this regard, it is interesting that Nhsl1b is expressed in the ventral, lateral and paraxial mesoderm, but not in the axial mesoderm, which displays a more directed and persistent migration, nor in the endoderm, which displays a random walk. During development, cells acquire specific fates and corresponding specific behaviours that are key to the proper morphogenesis of the embryo. How fate instructs cell behaviour remains surprisingly poorly understood. While a number of actin regulators have been shown to affect morphogenetic movements[13,40–46], very few have been identified as targets of cell fate inducers, showing tissue specific expression. Nhsl1b appears to be such a target and provides a new clue as to how cell fate can control cell behaviour and direct morphogenesis.

## Methods

### Zebrafish lines and husbandry
Embryos were obtained by natural spawning of AB and *Tg(tbx16:EGFP)*[47] adult fishes. Embryos were incubated at 28 °C and staged in hours post-fertilization (hpf) as described by Kimmel et al.[48]. All animal experimentation was conducted in accordance with the local ethics committee and approved by the Ethical Committee N°59 and the Ministère de l'Education Nationale, de l'Enseignement Supérieur et de la Recherche under the file number APAFIS#21670-2019073114516116 v2.

### Table 1 | Morpholinos sequences and concentrations

| Name | Sequence (5'-3') | Concentration |
|---|---|---|
| MO-1 Nhsl1b[25] | CGGGAAACGGCATTTTAAATCCTGT | 0.5 mM |
| MO-2 Nhsl1b | ATCCTGTTCAAATCTGAAGCGAGCA | 0.5 mM |
| MO Sox32[57] | CAGGGAGCATCCGGTCGAGATACAT | 0.3 mM |
| standard control | CCTCTTACCTCAGTTACAATTTATA | 0.5 mM |

### Table 2 | Primers for *nhsl1b-mNeongreen* cloning

| Primer | Sequence (5'-3') |
|---|---|
| Nhsl1b infusion F | GAGAGGCCTTGAATTCCTAACTGCTGCTCTGCTCACA |
| Nhsl1b infusion R | TCTTTTTGCAGGATCCATGCCGTTTCCCGAGAGAG |

### Table 3 | Primers for *nhsl1* in situ hybridisation probe synthesis

| Primer | Sequence (5'-3') |
|---|---|
| Nhsl1b F | TTGATTGCCACTCTCCAACAGT |
| Nhsl1b R | TAATACGACTCACTATAGGCAGGGGAGGAATTGTTTTGAAG |

### Zebrafish injection
Translation blocking morpholinos (Gene Tool LLC Philomath) were injected in 1-cell stage embryos with 2nL of injection volume. Following morpholinos and concentrations were used (Table 1):

For rescue experiments, *nhsl1b* was cloned in pCS2+ plasmid with In-Fusion Cloning Kit (Takara). Primer sequences are provided in Table 2. *nhsl1b* was then sub-cloned into a pCS2 + -mNeongreen plasmid in order to obtain a C-terminally tagged Nhsl1b-mNeongreen protein (Addgene plasmid # 233883). Plasmids were linearized with NotI and capped mRNAs were synthetized using the mMessage mMachine SP6 kit (Thermo Fischer). *nhsl1b* mRNA was injected in 1-cell stage embryos with 2nL of injection volume and 40 ng.µL$^{-1}$ for rescue experiments, 30 ng.µL$^{-1}$ for overexpression experiments. For the transplant experiments donor embryos were injected in one cell of 4-cell stage with a 1nL volume of injection and the following mRNAs concentrations: *lifeact-mCherry* (30 ng.µL$^{-1}$), *lifeact-mNeongreen* (30 ng µL), *nhsl1b-mNeongreen* (15 ng.µL$^{-1}$).

### Whole-mount in situ hybridization
Zebrafish embryos were fixed in 4% paraformaldehyde at 4 °C for 48 hpf. Whole-mount in situ hybridisations were performed according to standard procedures[49,50]. A 835 bp long *nhsl1b* probe was prepared by in vitro translation of a PCR amplified template. Used primers are indicated in Table 3. A sense probe of similar size was used as a control.

### CRISPR/Cas13d
Three different guide RNAs targeting *nhsl1b* (ENSDART00000062532.4) were prepared following Kushawah et al. protocol[32]. Targeted sequences can be found in the following table.

*cas13d* mRNA was synthetized from the plasmid pT3TS-RfxCas13d-HA # 141320 (addgene). *cas13d* mRNA and a mix of 3 guide RNAs targeting *nhsl1b* (Table 4) were injected in 1-cell stage embryos with 2 nL of injection volume and 600 ng.µL$^{-1}$ (gRNAs) and 300 ng.µL$^{-1}$ (*cas13d*) concentrations. *cas13d* mRNA was injected alone as a control, as performed in Kushawah et al.

### RT-qPCR
RNA was isolated from 9 hpf embryos injected with *cas13d* mRNAs with or without *nhsl1b* guide RNAs. Embryos were kept in RNAlater solution (Sigma Aldrich) and RNA was extracted using RNeasy kit (Qiagen) according to the manufacturer's instructions. Total RNA was processed for reverse transcription (RT) using AccuScript High-Fidelity First Strand cDNA Synthesis Kit (Agilent). A mix of anchored-Oligo(dT) and random primers (9-mers) was used to generate the cDNA. Real-time PCR reactions were carried out using SYBR green Master Mix on a CFX96 Real Time system (BIO-RAD). Primer sequences are provided in Table 5. Gene expression levels were determined by the $2 - \Delta\Delta CT$ method following normalisation to *cdk2ap* used as a reference gene. RT-qPCR experiments were repeated at least three times with independent biological samples; technical triplicates were run for all samples.

### Cell transplantations
To obtain mosaic embryos, labelled cells from a donor embryo were transplanted into a wild-type non-labelled host embryo[51,52]. Donor wild-type embryos were co-injected with *Lifeact-mCherry* or *Lifeact-mNeongreen* mRNA and either MO Control, MO Nhsl1b, *lacZ* or *nhsl1b* mRNA. Cells were collected at the margin of a late blastula (4 hpf) donor embryo and

### Table 4 | Sequences of guide RNAs targeting *nhsl1b*

| Name | Sequence (5'-3') |
|---|---|
| gRNA_Nhsl1b_1 | GCCGGTTGAGGGGGAGCGATGGGTTTCAAACCCCGACCAGTT |
| gRNA_Nhsl1b_2 | AGACTCAAGCTGGGCTAATCTCGGTTTCAAACCCCGACCAGTT |
| gRNA_Nhsl1b_3 | ACTCCTCCTTTGATCCGGCGTCTGTTTCAAACCCCGACCAGTT |

**Table 5 | Primer sequences for RT-qPCR**

| Primer | Sequence (5'-3') |
|---|---|
| nhsl1b F | CCCAAATCGGTTAAAATACCTGT |
| nhsl1b R | TTTGGCCTGACGGTGAAGAT |
| cdk2ap2_F | AGGATCTTGTGGCTCTTCTCCATCAC |
| cdk2ap2_R | TTTCACGGCTCATCTCCTCAATGAC |

transplanted at the margin of a host embryo at the same stage. At early gastrulation (6.5 hpf), embryos with transplanted cells in the lateral mesoderm were selected and mounted for imaging.

To look at the expression of *nhsl1b* in response to Nodal signalling and in endoderm, donor embryos were injected at the 4-cell stage with 1 nL of GFP mRNAs at 50 ng.µL$^{-1}$ and *ndr1* mRNAs at 20 ng.µL$^{-1}$ or *sox32* mRNAs at 25 ng.µL$^{-1}$. At the shield stage, a few green cells were transplanted from the margin to the animal pole of a host embryo. Embryos were fixed at mid-gastrulation (75% epiboly) stage and analysed by in situ hybridisation.

**Mesoderm ex vivo imaging**

Wild-type embryos at the 4-cell stage were injected in one cell with *acvr1ba\** mRNA (0.6 ng.µL$^{-1}$) and Sox32 morpholino (0.3 mM) to induce a mesodermal identity[53], together with *Lifeact-mCherry* (50 ng.µL$^{-1}$), and *Nhsl1b-mNeongreen* (15 ng.µL$^{-1}$) mRNAs to visualise F-actin and the subcellular localisation of Nhsl1b. At early gastrulation (60% epiboly), a group of about 200 labelled cells was manually dissected in dissection medium and dissociated in Ringer's without calcium solution. Clusters of fewer than 10 cells were transferred in a drop of culture medium to an E-cadherin coated glass bottom chamber. Clusters were incubated at 28 °C for 20 min to allow cells to attach to the glass coverslip and imaged using an inverted confocal microscope. Dissection Medium: 1X MMR + BSA 0.1% [10X MMR : 1 M NaCl, 20 mM KCl, 10 mM MgCl2, 20 mM CaCl2, 50 mM HEPES (pH 7.5)]. Culture Medium: 80% Leibovitz's L-15 medium (Thermofischer) diluted at 65% in water + 20% Embryo medium + 0.1% BSA + 125 mM HEPES (ph7.5) + 10 U/mL Penicillin/Streptomycin.

**Imaging**

Imaging of embryos for cell tracking and protrusion dynamics analysis was done under an upright TriM Scope II (La Vision Biotech) two-photon microscope equipped with an environmental chamber (Okolab) at 28 °C and a XLPLN25XWMP2 (Olympus) 25x water immersion objective. Labelled embryos were mounted laterally in 0.2% agarose in embryo at 6.5 hpf. Embryos were imaged every 1 or 2 min for 60 min. Imaging of mesodermal cells plated on a coverslip and high temporal resolution imaging of actin flows were done on an inverted TCS SP8 confocal microscope (Leica) equipped with environmental chamber (Life Imaging Services) at 28 °C using a HC PL APO 40x/1.10WCS2 objective (Leica). Whole embryo imaging was performed with a M205FCA stereomicroscope (Leica) and a MC170HD Camera (Leica). Images were processed in Fiji and Adobe Photoshop.

**Image analysis**

For cell tracking, *Tg(tbx16:egfp)* embryos were injected with *H2B-mCherry* (50 ng.µL$^{-1}$) mRNA at the 1-cell stage and mounted at 6.5 hpf. Nuclei were tracked in IMARIS (Bitplane) and further processed using Matlab (Math Works) to compute instant cell speed, directionality ratio, and directional autocorrelation[53,54]. For each tracked embryo, instant speed and persistence were computed for every cell at each time interval. Instant cell speed was calculated using a time step of 6 min over a total duration of one hour. Persistence was defined as the ratio between net displacement and total displacement, computed over 10-min intervals. Outliers were removed according to Chauvenet criterion. Mesoderm and epiboly progression were quantified on *Tg(tbx16:egfp)* embryos. Actin-rich protrusions were quantified on Lifeact-mCherry or Lifeact-mNeongreen expressing cells. Kymograph were generated from a maximum projection of a 50-pixel width line crossing the protrusion. The KymographClear FIJI macro toolset was used to separate backward and forward motions, through a Fourier transform based algorithm.

**Statistics and reproducibility**

Statistical analyses were performed in R (R Core Team, 2024). For independent observations, significance was calculated using Mann–Whitney's test or Kruskal–Wallis test followed up with Dunn's multiple comparison test. For non-independent observations (several measurements for each cell and several cells for each embryo) we used lme4[55] and nlme[56] to perform mixed effects analyses. As fixed effects we entered the treatment (Control, MO Nhsl1b, MO-2 Nhsl1lb), as random effects we had intercepts for embryos or cells. Visual inspection of residual plots did not reveal any obvious deviations from homoscedasticity or normality. *P*-values were obtained by likelihood ratio tests of the full model with the fixed effect against the model without the fixed effect. When multiple comparisons were performed, *p*-values were adjusted using the Benjamini Hochberg method. All data were modelled with linear models, except directional autocorrelation which was fit to an exponential decay with a plateau as shown below:

$$A = \left(1 - A_{min}\right)e^{-\frac{t}{\tau}} + A_{min}$$

where $A$ is the autocorrelation, t the time interval, $A_{min}$ the plateau and $\tau$ the time constant of decay.

In all figures, ns: *p*-value ≥ 0.05; *$p < 0.05$; **$p < 0.01$; ***$p < 0.001$. Box-plots show min, max and median. Violin plots show median and quartiles.

**Reporting summary**

Further information on research design is available in the Nature Portfolio Reporting Summary linked to this article.

**Data availability**

The images from the figures, the numerical source data behind the graphs and the corresponding statistical analysis are available at https://doi.org/10.6084/m9.figshare.28071539. Further raw data are available from the corresponding author upon reasonable request.

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

## Acknowledgements

We thank Emilie Menant for fish care; P. Mahou and the Polytechnique Bioimaging Facility for imaging on their equipment supported by Region Ile-de-France (interDIM) and Agence Nationale de la Recherche (ANR-11-EQPX-0029 Morphoscope2, ANR-10-INBS-04 France BioImaging). We thank Sylvie Rétaux for interest and support (ANR CAVEMOM for S.R.). This work was supported by grants ANR-18-CE13-0024, ANR-20-CE13-0016-03 and an Institut Polytechnique de Paris, E4H Serge Schoen New Synergies Grant for N.D. S.E. was supported by the European Union's Horizon 2020 programme under the Marie Skłodowska-Curie grant agreement No 840201.

## Author contributions

S.E. and N.D. conceived experiments, which were performed by S.E. Y.H. contributed to the MO-2 Nhsl1b experiments. J.T.P. performed the double in situ hybridisation experiment. A.E. developed the protocol to explant mesodermal cells. L.M. provided help with cloning of Nhsl1b and Nhsl1b-mNeongreen. AI provided helped with the in situ hybridisation experiments. S.E. analysed data. S.E. and N.D. wrote the manuscript. N.D. secured funding.

## Competing interests

The authors declare no competing interests.
