## [Transparent Peer Review file · Communications Biology]

Nance-Horan-Syndrome-like 1b controls mesodermal cell migration by regulating protrusion and actin dynamics during zebrafish gastrulation

Corresponding Author: Dr Nicolas David

Version 0:

Reviewer comments:

Reviewer #1

(Remarks to the Author)

Escot et al describe the function of NHSL1b in mesoderm migration during zebrafish gastrulation. They report that knockdown and overexpression of NHSL1b both causes reduced mesoderm migration. Both knockdown as well as overexpression decreases cell speed and persistence. However, NHSL1b knockdown reduces protrusion lifetime and length but not protrusion speed whereas NHSL1b overexpression increases protrusion lifetime and length. NHSL1b knockdown was then shown to increase both actin retrograde flow and F-actin assembly rates. The authors compared their results to the results of two previous studies which reported NHSL1 function in mammalian cultured cells in 2D migration assays. Here in the in vivo zebrafish system the authors observed some differences (overall speed and persistence reduces whereas in the 2D system it increases) but also some similarities (exact levels of NHSL1 appear important as knockdown and overexpression have similar effects). They correctly speculate that the differences may be due to differences in the type of migration tested: random migration versus directed migration. Overall, this is an interesting study which reveals more details on how NHSL1b may control different types of migration or in different contexts. I am happy to support the publication of this study if my concerns have been addressed.

Major concerns:

1. It is good to see that in Figure 1 you verified that the effect of knocking down NHSL1b is not due to off-target effects of the morpholino used. However, in Fig. 3, 6, 7: you cannot just assume that because a morpholino did not show off target effects in one experiment that it will not show off target effects in other measurements. You need to repeat these experiments with the second MO as well or alternatively rescue your phenotypes to be sure that the measured effects are not caused by off target effects.

2. The authors should describe the statistical test used in their experiments better: I assume from their statement in the figure legend "Linear mixed-effect models." And the stats section in the methods: "ANOVA on mixed-effect models for nested measurements (several measurements for each cell, several cells for each embryo, and several embryos per experiments)." Do the authors mean repeated measures ANOVA? This would not be the correct test here as the same embryos were not repeatedly measured. Or do they mean Nested ANOVA? Also, this would not be appropriate as for each repeat several embryos were treated in parallel with different MOs. Thus, One-Way ANOVA with Tukeys or here more likely the Kruskal-Wallis test would be more appropriate as it does not look like a normally distributed dataset. The appropriate test used should be stated in the figure legend with the actual P values.

Detailed comments:

Fig 1BC; Fig. 2: How often was the experiment repeated? Please state how many independent experiments were performed in the figure legend.

Fig 3c: "observed a reduction of persistence in nhsl1b knocked-down cells" please rephrase as you did not observe a significant reduction in persistence. Writing "mixed model" in the figure legend is not sufficient and rather will confuse the reader. Please use the correct stats for each experiment and state the correct stats used for each panel including p values in the figure legend. Without this, the reader cannot judge whether your statements are backed by the data. For 3c this would be Mann-Whitney test.

Fig. 3d: Which statistical test did you use to do stats on the direction autocorrelation?

Fig 4,6,7: Please use correct stats and state which statistical test you used for each panel. See above.

Reviewer #2

(Remarks to the Author)

Summary of the manuscript

The manuscript Escot et al. reports a role for Nhs1b, a relatively understudied regulator of the Scar/WAVE complex, in mesodermal cell migration during zebrafish gastrulation. The authors confirm previous reports that nhs1b expression is induced by Nodal signaling and extend these observations to show that expression is primarily restricted to the mesodermal lineage. The authors then perform gain and loss of function experiments to probe the function of nhs1b in mesodermal cells. They show that both loss of nhs1b (via MO and CRISPR/Cas-13 knockdown) or overexpression negatively impact migration of mesodermal cells towards the animal pole, reducing both migration speed and directionality. The authors go on to explore the underlying mechanisms contributing to these impaired migration phenotypes at the single-cell level. By expressing a fluorescent fusion protein, they show that nhs1b is localized to the tips of leading edge protrusions as well as sites of cell-cell contact. They also show that loss of nhs1b results in reduced protrusion length and lifetime and increased rates of F-actin assembly and retrograde flow. In contrast, nhs1b overexpression increases the length and lifetime of protrusions, leading the authors to conclude that nhs1b is required to “fine tune” actin and protrusion dynamics to promote efficient migration towards the animal pole.

Overall impression of the work:

This manuscript is an important advancement in our understanding of how Nhs1b regulates cell migration. Not much is known this particular actin regulator, and this manuscript is significant in that it examines nhs1b function in an in vivo and cell type-specific context, an approach that is often missing in cytoskeletal biology papers. A notable strength of the manuscript is their rigorous use of quantitative approaches to analyze the effects of nhs1b on cell migration, protrusion, and actin dynamics. In general, the experiments are adequately designed to justify the authors' conclusions. However, the manuscript would be strengthened if the authors could address the specific comments in the following section.

Specific Comments:

1. The authors conclude that nhs1b expression is mesoderm-specific, based in part on whole-mount in situ hybridization presented in Fig. 1a. However, from the images provided, it is not clear whether nhs1b expression is exclusive to the mesoderm. For example, the intense dorsal staining at mid-gastrulation appears to overlap with the location of the dorsal forerunner cells/KV, which are not mesodermal in origin (Warga and Kane, 2018, DOI: 10.1002/dvdy.24657). A recommendation would be to co-label with a mesodermal marker or perform sectioning to more definitively show that nhs1b expression is restricted to the mesoderm.
2. Mesoderm-specific nhs1b expression was also based on the transplantation experiments presented in Fig. 1B. in which cells overexpressing ndr1 express nhs1b but cells overexpressing sox32 do not. Again, their claims would be strengthened by performing double in situs with nhs1b and mesodermal and endodermal markers to show that nhs1b expression overlaps with the former and is mutually exclusive with the latter. If these double in situ experiments are not feasible, the authors should at least qualify their conclusions based on Fig. 1a and 1b.
3. It is arguable whether measuring progression of the mesoderm front relative to the margin is preferable to measuring from the animal pole. As the authors mention, the location of the margin is constantly shifting due to epiboly, which complicates their analysis and potentially under-reports the effects of nhs1b loss and gain of function. Fig. S1 shows that measuring from the animal pole is a valid, and potentially less problematic, alternative. Would it be possible to use this approach for all figures in the main text? The authors state that measuring from the animal pole is less accurate because of the sphericity of the embryos, but that would presumably be an issue for all points on the embryo.
4. In Fig. 2b, the authors show that the MO-induced migration defects can be rescued by co-injection with nhs1b mRNA. However, this interpretation is complicated by data in Fig. 4 showing that overexpression of nhs1b also affects mesodermal migration. In fact, the data points in MO+mRNA “rescues” appear much more variable than control, which may be due to confounding effects of mRNA overexpression. This issue could be clarified by performing a dose response curve with increasing amounts of nhs1b mRNA. It may also be interesting to see whether the effects on migration occur progressively or at certain threshold.
5. In the discussion, the authors note that their observation that nhs1b knockdown increases retrograde flow but decreases protrusion lifetime and migration persistence is in contrast to a proposed “universal law” that increased retrograde flow increases persistence. However, the authors might want to note their observations are consistent with other reports linking slowed retrograde flow with increased protrusion persistence, for example, Lim et al., 2010 (DOI: 10.1016/j.yexcr.2010.04.011) and Yang et al., 2012 (DOI: 10.1083/jcb.201111052).
6. In all figures, the asterisks denoting statistical significance should be defined in the figure legends (p value and test used).
7. It would be helpful to add axis labels to the graphs in Fig. 3e and 3f.

Version 1:

Reviewer comments:

Reviewer #1

(Remarks to the Author)

The authors have addressed all my concerns appropriately and I am now happy to support publication. Overall this is now a very well done study - congratulations!

Reviewer #2

(Remarks to the Author)

Summary of the manuscript

The revised manuscript by Escot et al. reports a role for *nhsl1b*, a relatively understudied regulator of the Scar/WAVE complex, in mesodermal cell migration during zebrafish gastrulation. Using *in vivo* live imaging techniques, the authors show that *nhsl1b* affects cell migration speed and directionality as well as actin and protrusion dynamics in mesodermal cells. Some of these effects are in contrast to previous reports on NHSL1 function, suggesting that this actin regulator can act in a context-specific manner.

Overall impression of the work:

The authors have sufficiently addressed the reviewer comments. The manuscript is much improved especially with respect to additional information provided on statistical analyses used and confirmation and clarification of the mesoderm-specific expression of *nhsl1b*. Minor suggestions are provided below to improve readability of the manuscript.

Specific comments

1. The addition of double fluorescent *in situ* to Fig. 1 are highly appreciated and strengthen the authors' conclusions that *nhsl1b* is expressed in the mesoderm. I suggest displaying the panels in Fig. 1D in a green/magenta color scheme (as the authors did for all other figures) rather than green/red to improve accessibility.
2. The dose-response analysis for *nhsl1b* mRNA injection provided in the rebuttal letter is much appreciated. If possible, I encourage the authors to include these data in the manuscript (perhaps as supplemental information) as it provides important controls for the rescue experiments and nicely demonstrates the authors' conclusions that *nhsl1b* levels need to be finely tuned.
3. The addition of axis labels and a schematic to Fig. 3E–F has improved readability of this figure. I encourage the authors to also include brief axis definitions to the figure legend (e.g., "y-axis represents distance along the animal-vegetal axis, x-axis represents distance along the dorsal-ventral axis").

Response to reviewer comments

We sincerely thank the reviewers for their insightful and constructive feedback, which has helped us to improve the manuscript. In response to their comments, we have performed additional experiments and analyses, resulting in the addition of one new panel (Fig 1D) and the updating of five sets of panels (Fig 3B-D; Fig 6B-C), included at the end of this document. We have also revised the text and provide a new version with clear indications of changes. Below, is a detailed point-by-point response, with our responses highlighted in blue. We hope that the reviewers and editor will find the revised manuscript suitable for publication in *Communications Biology*.

Reviewers' comments:

Reviewer #1 (Remarks to the Author):

Escot et al describe the function of NHSL1b in mesoderm migration during zebrafish gastrulation. They report that knockdown and overexpression of NHSL1b both causes reduced mesoderm migration. Both knockdown as well as overexpression decreases cell speed and persistence. However, NHSL1b knockdown reduces protrusion lifetime and length but not protrusion speed whereas NHSL1b overexpression increases protrusion lifetime and length. NHSL1b knockdown was then shown to increase both actin retrograde flow and F-actin assembly rates. The authors compared their results to the results of two previous studies which reported NHSL1 function in mammalian cultured cells in 2D migration assays. Here in the *in vivo* zebrafish system the authors observed some differences (overall speed and persistence reduces whereas in the 2D system it increases) but also some similarities (exact levels of NHSL1 appear important as knockdown and overexpression have similar effects). They correctly speculate that the differences may be due to differences in the type of migration tested: random migration versus directed migration. Overall, this is an interesting study which reveals more details on how NHSL1b may control different types of migration or in different contexts. I am happy to support the publication of this study if my concerns have been addressed.

Major concerns:

1. It is good to see that in Figure 1 you verified that the effect of knocking down NHSL1b is not due to off-target effects of the morpholino used. However, in Fig. 3, 6, 7: you cannot just assume that because a morpholino did not show off target effects in one experiment that it will not show off target effects in other measurements. You need to repeat these experiments with the second MO as well or alternatively rescue your phenotypes to be sure that the measured effects are not caused by off target effects.

We thank the reviewer for raising this important point. First, we would like to respectfully point that, owing to the difficulty of many of the *in vivo* experiments, it is fairly common in the zebrafish literature to validate a knock-down approach on one phenotype, and use it in the following experiments, or even in other studies (1–3). We also believe that the observation that overexpression of *Nhsl1b* has opposite effect to its knockdown on protrusion lifetime and length argues against these phenotypes being off target effects. Nevertheless, we agree with the reviewer that this cannot be ascertained, and have therefore, as suggested, used the second independent morpholino. Doing so, we confirmed the specificity of the effect on cell

speed, directionality and autocorrelation (Fig 3), as well as the effect on protrusion length and lifetime (Fig 6). Owing to the difficulty of these experiments and to our confidence in the specificity of the protrusion phenotype (2 morpholinos + opposite effect with over-expression), we have not analysed actin dynamics with the second morpholino (Fig 7). We believe the added data address the reviewer's concern and remain open to further analyses if the reviewer deems it necessary.

1. Roussigné, M. et al. Left/right asymmetric collective migration of parapineal cells is mediated by focal FGF signaling activity in leading cells. *Proc. Natl. Acad. Sci. USA* 115, E9812–E9821 (2018).
2. Lundin, V. et al. YAP regulates hematopoietic stem cell formation in response to the biomechanical forces of blood flow. *Dev. Cell* 52, 446–460.e5 (2020).
3. Balaraju, A. K., Hu, B., Rodriguez, J. J., Murry, M. & Lin, F. Glypican 4 regulates planar cell polarity of endoderm cells by controlling the localization of Cadherin 2. *Development* (2021). doi:10.1242/dev.199421

2. The authors should describe the statistical test used in their experiments better: I assume from their statement in the figure legend “Linear mixed-effect models.” And the stats section in the methods: “ANOVA on mixed-effect models for nested measurements (several measurements for each cell, several cells for each embryo, and several embryos per experiments).” Do the authors mean repeated measures ANOVA? This would not be the correct test here as the same embryos were not repeatedly measured. Or do they mean Nested ANOVA? Also, this would not be appropriate as for each repeat several embryos were treated in parallel with different MOs. Thus, One-Way ANOVA with Tukeys or here more likely the Kruskal-Wallis test would be more appropriate as it does not look like a normally distributed dataset. The appropriate test used should be stated in the figure legend with the actual P values.

We thank the reviewer for bringing up this important point regarding our choice of statistical analysis. We apologize if our statistical explanations were misleading or unclear.

We agree with the reviewer that the Kruskal-Wallis test, followed by Dunn's multiple comparison test for analyses involving more than two conditions, or Mann-Whitney for analyses involving two conditions, are the most suitable approaches for analysing mesoderm progression. We have therefore used these tests in Figure 2B, 4B, S1 and S2, and updated the figure legends and the Methods section.

For Figure 3B-D, 4C-D, 6, and 7 we have performed, as suggested, Kruskal-Wallis or Mann-Whitney tests (see table 1 below), which confirm the significance of the observed effects. However, one of the key assumptions of Kruskal-Wallis and Mann-Whitney tests is that all observations are independent, which they are not here, as multiple cells are measured in one embryo, or multiple measurements are performed on the same cell over time. Though not yet very common in cell biology, mixed effects models appear as powerful tools for analysing such datasets (1–3). We used them here, as described in (4). In short, two models are used, the first one takes into account a fixed effect (treatment) and a random effect (embryos or cells). The second model only takes into account the random effect. The two models are compared using an ANOVA, to see if taking into account the treatment significantly improves the model (i.e. the treatment has a significant effect). When multiple comparisons were performed (second

morpholino), p-values were adjusted using the Benjamini Hochberg method. We believe this approach is better suited than Kruskal-Wallis tests as it copes with the non-independent nature of the data. We have therefore used it in the manuscript, but could switch to Kruskal-Wallis if the reviewer believes this is a better approach.

We apologize that our use of mixed effects models was not properly explained in the manuscript. We have modified the methods section, and have updated the figure legend to explicitly include the statistical test used and the obtained p-values.

1. Schielzeth H, Dingemanse NJ, Nakagawa S, Westneat DF, Alague H, Teplitsky C, et al. Robustness of linear mixed-effects models to violations of distributional assumptions. *Methods Ecol Evol.* 2020 Jun 12;
2. Yu Z, Guindani M, Grieco SF, Chen L, Holmes TC, Xu X. Beyond t test and ANOVA: applications of mixed-effects models for more rigorous statistical analysis in neuroscience research. *Neuron.* 2022 Jan 5;110(1):21–35.
3. Bolker BM, Brooks ME, Clark CJ, Geange SW, Poulsen JR, Stevens MHH, et al. Generalized linear mixed models: a practical guide for ecology and evolution. *Trends Ecol Evol (Amst).* 2009 Mar;24(3):127–135.
4. Winter B. Linear models and linear mixed effects models in R with linguistic applications. *arXiv.* 2013;

Figure 2: *nhs1b* knockdown affects lateral mesoderm migration.

	Kruskal-Wallis (Dunn's multiple comparison)
Mesoderm progression	
MO Control vs MO Nhs1b	0,0093 **
MO Control vs MO-2 Nhs1b	0,0002 ***
MO Control vs MO-2 Nhs1b + Nhs1b	0,6481 ns
MO-2 Nhs1b vs MO-2 Nhs1b + Nhs1b	0,0218 *

Figure 3: *nhs1b* knockdown reduces cell speed and persistence.

	Linear Mixed Effects Models. Adjusted p-values	Kruskal-Wallis (Dunn's multiple comparison)
Directionality ratio		
MO Control vs MO Nhs1b	0,03019 *	<0,0001 * ***
MO Control vs MO-2 Nhs1b	0,03019 *	<0,0001 * ***
Instant Speed		
MO Control vs MO Nhs1b	0.01003614 *	<0,0001 * ***
MO Control vs MO-2 Nhs1b	0.01313286 *	<0,0001 * ***
Autocorrelation		
MO Control vs MO Nhs1b	0,045 *	
MO Control vs MO-2 Nhs1b	0,005411 **	

Figure 4: *nhs1b* overexpression affects lateral mesodermal migration, reducing cell speed and persistence.

	Mann Whitney	Mann Whitney
Mesoderm progression		
Control vs Nhs1b	0,0075 **	
Directionality ratio		
Control vs Nhs1b	0.01117 *	<0,0001 * ***
Instant speed		
Control vs Nhs1b	0.00299 **	<0,0001 * ***
Autocorrelation		
Control vs Nhs1b	0,03589163 *	

Figure 6: Nhs1b regulates protrusion dynamics.

	Linear Mixed Effects Models. Adjusted p-values		Mann Whitney
Protrusions length			
MO Control vs MO Nhs1b	0,000132 ***		<0,0001 * ***
MO Control vs MO-2 Nhs1b	4,89E-06 ***		<0,0001 * ***
Protrusions duration			
MO Control vs MO Nhs1b	0,0309 *		0,0032 ** ***
MO Control vs MO-2 Nhs1b	0,00111 ***		<0,0001 * ***
	Linear Mixed Effects Models		Mann Whitney
Protrusions length			
Control vs Nhs1b	0.0001257 ***		<0,0001 * ***
Protrusions duration			
Control vs Nhs1b	0.009365 **		0,0007 ***

Figure 7: *nhs1b* knockdown increases F-actin retrograde flow and assembly rate.

	Linear Mixed Effects Models		Mann Whitney
F-actin assembly rate			
MO Control vs MO Nhs1b	0,02481 *		0,0194 *
Retrograde flow			
MO Control vs MO Nhs1b	0.005066 **		0,0002 ***
Protrusion extension			
MO Control vs MO Nhs1b	0.8362 ns		0,0804 ns

Figure S1: Epiboly and mesoderm front progression in *nhs1b* knockdown.

	Mann Whitney
Epiboly progression	
MO Control vs MO Nhs1b	0,6947 ns
Mesoderm front progression	
MO Control vs MO Nhs1b	0,0028 **
Epiboly progression	
Control vs Nhs1b	0,0934 ns
Mesoderm front progression	
Control vs Nhs1b	0,0312 *

Figure S2: Nhs1b knockdown using CRISPR/Cas13d.

	Mann Whitney
RT-qPCR	
Cas13d vs Cas13d + gNhs1b	0,0003 ***
Mesoderm progression	
Cas13d vs Cas13d + gNhs1b	0,0303 *

Table 1: Statistical tests and their corresponding p values

Detailed comments:

Fig 1BC; Fig. 2: How often was the experiment repeated? Please state how many independent experiments were performed in the figure legend.

We have updated the figure legends to include the number of independent experiments.

Fig 3c: “observed a reduction of persistence in nhsl1b knocked-down cells” please rephrase as you did not observe a significant reduction in persistence. Writing “mixed model” in the figure legend is not sufficient and rather will confuse the reader. Please use the correct stats for each experiment and state the correct stats used for each panel including p values in the figure legend. Without this, the reader cannot judge whether your statements are backed by the data. For 3c this would be Mann-Whitney test.

We fully agree with the reviewer that Fig. 3C did not show a significant effect on persistence, and that the sentence was therefore inaccurate. While revising the manuscript, we have performed additional experiments, now showing a significant reduction of persistence, with both morpholinos.

Regarding the statistical tests, as discussed before, we need to consider the non-independence of measures. We therefore used mixed effects models, linear for directionality ratio, non-linear for directional autocorrelation (see below).

Additionally, the graph in 3C has been revised to better reflect the statistical analysis, showing the directionality ratio of all the cells, in addition to the mean per embryo. We have updated the figure legend for 3C to specify the use of LME models and included the corresponding p-value.

Fig. 3d: Which statistical test did you use to do stats on the direction autocorrelation?

As for the other measures, we have to consider their non-independence, and therefore used mixed effects models. However, directional autocorrelation decreases exponentially over time, and cannot be modelled using linear models. To model directional autocorrelation, we used the following exponential model:

$$A = (1 - A_{min}) * e^{-\frac{t}{\tau}} + A_{min}$$

where A is the autocorrelation, t the time interval, A_{min} the plateau and τ the time constant of decay. As for the other data, we compared two models, one with fixed (treatment) and random (embryos) effects on both A_{min} and τ , and one with random effects only.

This approach was designed for these autocorrelation data by a professional statistician (Marc Lavielle, researcher at the Center for Applied Maths, Ecole Polytechnique; now retired).

Methods and the figure legend were corrected to explain this and provide the p-values.

Fig 4,6,7: Please use correct stats and state which statistical test you used for each panel. See above.

As described above, we believe we have used correct statistical approaches, and apologize for not having presented them clearly in the initial manuscript. We have now updated the methods and figure legends for Figures 4, 6, and 7 as well as 2, 3, S1 and S2 to include the statistical tests used for each panel, along with the corresponding p-values.

If the reviewer feels another statistical approach would be preferable, we would be happy to follow his recommendations.

The statistical paragraph in the methods now reads:

“Statistical analyses were performed in R (R Core Team, 2024). For independent observations, significance was calculated using Mann-Whitney’s test or Kruskal-Wallis test followed up with Dunn’s multiple comparison test. For non-independent observations (several measurements for each cell and several cells for each embryo) we used lme456 and nlme57 to perform mixed effects analyses. As fixed effects we entered the treatment (Control, MO Nhs1b, MO-2 Nhs1b), as random effects we had intercepts for embryos or cells. Visual inspection of residual plots did not reveal any obvious deviations from homoscedasticity or normality. P-values were obtained by likelihood ratio tests of the full model with the fixed effect against the model without the fixed effect. When multiple comparisons were performed, p-values were adjusted using the Benjamini Hochberg method. All data were modelled with linear models, except directional autocorrelation which was fit to an exponential decay with a plateau as shown below:

$$A = (1 - A_{min}) * e^{-\frac{t}{\tau}} + A_{min}$$

where A is the autocorrelation, t the time interval, Amin the plateau and τ the time constant of decay.

In all figures, ns: p-value ≥0.05; *: p<0.05; **: p<0.01; ***: p<0.001. Box-plots show min, max and median. Violin plots show median and quartiles.”

Reviewer #2 (Remarks to the Author):

Summary of the manuscript

The manuscript Escot et al. reports a role for Nhs1b, a relatively understudied regulator of the Scar/WAVE complex, in mesodermal cell migration during zebrafish gastrulation. The authors confirm previous reports that nhs1b expression is induced by Nodal signaling and extend these observations to show that expression is primarily restricted to the mesodermal lineage. The authors then perform gain and loss of function experiments to probe the function of nhs1b in mesodermal cells. They show that both loss of nhs1b (via MO and CRISPR/Cas-13 knockdown) or overexpression negatively impact migration of mesodermal cells towards

the animal pole, reducing both migration speed and directionality. The authors go on to explore the underlying mechanisms contributing to these impaired migration phenotypes at the single-cell level. By expressing a fluorescent fusion protein, they show that *nhs1b* is localized to the tips of leading edge protrusions as well as sites of cell-cell contact. They also show that loss of *nhs1b* results in reduced protrusion length and lifetime and increased rates of F-actin assembly and retrograde flow. In contrast, *nhs1b* overexpression increases the length and lifetime of protrusions, leading the authors to conclude that *nhs1b* is required to “fine tune” actin and protrusion dynamics to promote efficient migration towards the animal pole.

Overall impression of the work:

This manuscript is an important advancement in our understanding of how *Nhs1b* regulates cell migration. Not much is known this particular actin regulator, and this manuscript is significant in that it examines *nhs1b* function in an *in vivo* and cell type-specific context, an approach that is often missing in cytoskeletal biology papers. A notable strength of the manuscript is their rigorous use of quantitative approaches to analyze the effects of *nhs1b* on cell migration, protrusion, and actin dynamics. In general, the experiments are adequately designed to justify the authors’ conclusions. However, the manuscript would be strengthened if the authors could address the specific comments in the following section.

Specific Comments:

1. The authors conclude that *nhs1b* expression is mesoderm-specific, based in part on whole-mount *in situ* hybridization presented in Fig. 1a. However, from the images provided, it is not clear whether *nhs1b* expression is exclusive to the mesoderm. For example, the intense dorsal staining at mid-gastrulation appears to overlap with the location of the dorsal forerunner cells/KV, which are not mesodermal in origin (Warga and Kane, 2018, DOI: 10.1002/dvdy.24657). A recommendation would be to co-label with a mesodermal marker or perform sectioning to more definitively show that *nhs1b* expression is restricted to the mesoderm.

Please see below.

2. Mesoderm-specific *nhs1b* expression was also based on the transplantation experiments presented in Fig. 1B. in which cells overexpressing *ndr1* express *nhs1b* but cells overexpressing *sox32* do not. Again, their claims would be strengthened by performing double *in situ* with *nhs1b* and mesodermal and endodermal markers to show that *nhs1b* expression overlaps with the former and is mutually exclusive with the latter. If these double *in situ* experiments are not feasible, the authors should at least qualify their conclusions based on Fig. 1a and 1b.

We thank the reviewer and appreciate the suggestion to validate the mesoderm-specific expression of *nhs1b*. To address this concern, we performed fluorescent double *in situ* hybridisation experiments. These proved to be particularly challenging due to the low expression levels of *nhs1b*. To overcome this difficulty, we collaborated with S. Retaux’s team, now co-authors of this manuscript, who have a strong expertise in optimizing fluorescent *in situ* hybridisations. Still, *nhs1b* signal in fluorescent *in situ* hybridisation is very low and only clearly detectable at the dorsal margin. Using double fluorescent *in situ* hybridisation for *sox17*

and *nhs1b*, we observed that *nhs1b* expression does not overlap with the *sox17*-positive dorsal forerunner cells. We have included these results in Figure 1.

Due to the impossibility of obtaining clear fluorescent staining outside the dorsal margin, we could not perform the suggested co-labelling with a mesodermal marker, either in whole embryos or in transplantation experiments. This impossibility had actually motivated the transplantation experiments presented in Figure 1B, which strongly suggest that *nhs1b* expression is mesoderm-specific.

To support this idea, we have examined available scRNAseq data (Danicell; Sur, A. et al. (2023). Dev. Cell 10.1016/j.devcel.2023.11.001). When analysing blastomeres (all blastomeres or early gastrulation blastomeres), the gene with the highest expression correlation with *nhs1b* is *tbxta*. There is no correlation with *sox17* or *sox32*.

Gene name	Correlation coefficient
nhs1b	1
tbxta	0.208
wnt11f2	0.190
arl4ab	0.170
rasgef1ba	0.159
...	
sox32	-0.014
sox17	-0.022

Taken together, we believe that these results point to a mesoderm specific expression of *nhs1b*. However, we agree that these are not absolute demonstration. To avoid any overstatement, we have revised the conclusion to read: “These results suggest that *nsh1b* expression during gastrulation is exclusive to mesodermal cells.”

Full added text (line 114 – line 122):

“Consistently, we analysed available scRNAseq data³⁰, and observed that, at the onset of gastrulation, the gene with the most correlated expression profile to *nhs1b* is *tbxta* ($r = 0.21$), a mesodermal marker, while there was no correlation to endodermal markers (*sox32*, $r = -0.01$; *sox17*, $r = -0.02$). Finally, we carefully analysed the dorsal marginal expression, which could overlap with the region containing forerunner cells, precursors of Kupffer’s vesicle, which are not mesodermal³¹. We performed double in situ hybridisation for *nhs1b* and *sox17* at mid gastrulation (75% epiboly) and bud stage (Fig. 1D) and observed that *sox17*-positive dorsal forerunner cells do not express *nhs1b*. These results suggest that *nsh1b* expression during gastrulation is exclusive to mesodermal cells.”

3. It is arguable whether measuring progression of the mesoderm front relative to the margin is preferable to measuring from the animal pole. As the authors mention, the location of the margin is constantly shifting due to epiboly, which complicates their analysis and potentially

under-reports the effects of *nhs11b* loss and gain of function. Fig. S1 shows that measuring from the animal pole is a valid, and potentially less problematic, alternative. Would it be possible to use this approach for all figures in the main text? The authors state that measuring from the animal pole is less accurate because of the sphericity of the embryos, but that would presumably be an issue for all points on the embryo.

We agree that measuring the mesodermal spread from the animal pole could be a valid alternative, as shown in Figure S1, and we appreciate the suggestion to apply this approach to all figures in the main text. As the reviewer points out, embryo sphericity is an issue for all points on the embryo. To limit the effect of sphericity, the distance to be measured should be as parallel as possible to the imaging plane. The problem with measuring from the animal pole stems from the fact that the animal pole cannot be identified. The only way to accurately position the animal pole is to use the margin as a reference, and mount the embryos with the margin perpendicular to the imaging plane (Figure A). In this position, the animal pole can be identified as the “top” of the embryo. However, in this position, the measured distance between the margin and the front (d_2) is close to the actual distance (blue arrow), whereas the measurement of the distance between the animal pole and the front is strongly affected by sphericity, and very sensitive to any imprecision in mounting.

Figure A: Schematic of mesoderm progression measurements.

This is why we chose to measure the margin to mesoderm front distance. While we agree that epiboly movements may lead to under-report the effect of *nhs11b* (though not drastically as suggested by Fig. S1), we prefer to stick to this accurate measure between two visible reference points, rather than risking including inaccurate measures.

4. In Fig. 2b, the authors show that the MO-induced migration defects can be rescued by co-injection with *nhs11b* mRNA. However, this interpretation is complicated by data in Fig. 4 showing that overexpression of *nhs11b* also affects mesodermal migration. In fact, the data

points in MO+mRNA “rescues” appear much more variable than control, which may be due to confounding effects of mRNA overexpression. This issue could be clarified by performing a dose response curve with increasing amounts of *nhs1b* mRNA. It may also be interesting to see whether the effects on migration occur progressively or at certain threshold.

We agree with the reviewer that the rescue data show more variability than control (standard deviation of rescue = 28,03; sd control = 17,98), however this variability is already present in the MO alone experiments (sd MO-1 = 26.10; sd MO-2 = 27.11), suggesting it is not related to the rescue itself. We nevertheless agree that, as the overexpression itself induces migration defects (Fig. 4), the rescue experiments are sensitive to the amounts of mRNA. As suggested, we performed an overexpression dose-response analysis (Figure B). Low doses (5 and 15 ng/uL) had no significant effect of mesoderm progression, 30 ng/uL being the lowest dose inducing an effect. We also performed rescue experiments with different concentrations of *nhs1b* mRNA (Figure C). Low concentrations (5 and 15 ng/uL) had no significant effect. A higher concentration (50 ng/uL) seems to have the opposite effect. The dose we used for the rescue is in between (40 ng/uL). It is slightly higher than the dose used in overexpression, which makes sense as in rescue experiment, there is no endogenous contribution. Overall, these analyses are in good agreement our conclusion that *nhs1b* levels need to be finely tuned.

As we have not performed three independent experiments for each of the doses, we preferred not to include these results in the manuscript, but are willing to do so if the reviewer thinks it is relevant.

Figure B: Dose-response analysis of *Nhs1b* mRNA on mesodermal migration. Quantification of the lateral mesoderm progression in control and embryos injected with different concentrations of *nhs1b* mRNA. Kruskal-Wallis test followed by Dunn's multiple comparison test, adjusted *p*-values: Control vs *Nhs1b* 5ng/µL >0.9999; Control vs *Nhs1b* 15ng/µL >0.9999; Control vs *Nhs1b* 30ng/µL 0.0269.

Figure C: Rescue of MO-2 Nhs11b phenotype on mesoderm progression by co-injecting different concentrations of nhs11b mRNA. Kruskal-Wallis test followed by Dunn's multiple comparison test, adjusted p-values: MO Control vs MO-2 Nhs11b: <0,0001 ****; MO Control vs MO-2 Nhs11b + 50ng/μL Nhs11b: 0,0364 *; MO-2 Nhs11b vs MO-2 nhs11b + 40ng/μL Nhs11b: 0,0208 *.

5. In the discussion, the authors note that their observation that nhs11b knockdown increases retrograde flow but decreases protrusion lifetime and migration persistence is in contrast to a proposed “universal law” that increased retrograde flow increases persistence. However, the authors might want to note their observations are consistent with other reports linking slowed retrograde flow with increased protrusion persistence, for example, Lim et al., 2010 (DOI: 10.1016/j.yexcr.2010.04.011) and Yang et al., 2012 (DOI: 10.1083/jcb.201111052).

We thank the reviewer for the suggestion. We have included this point in the discussion. The paragraph now reads:

“These in vitro / in vivo differences highlight the importance of analysing cell migration regulators in their physiological context, as the complex and partially understood feedbacks that regulate actin assembly and migration persistence may lead to different outcomes depending on the cell state and environment. We observed that in Nhs11b knockdown, which leads to a reduction in protrusion length and in cell persistence there is an increase of F-actin retrograde flow. While these observations are consistent with some previous in vitro studies that have observed an increase of F-actin retrograde flow in non-persistently protruding cells 36,37 they also contrast with the proposed law that increased retrograde flow stabilises cell polarity and increases persistence38. This suggests that the relationship between retrograde flow and migration persistence is more nuanced than previously appreciated and reinforces the need to consider cellular and environmental context in understanding migration regulation.”

6. In all figures, the asterisks denoting statistical significance should be defined in the figure legends (p value and test used).

We have updated all figures legends to define the statistical tests used and their corresponding p-values.

7. It would be helpful to add axis labels to the graphs in Fig. 3e and 3f.

We have added axis labels to the graphs in Figures 3E and 3F. Additionally, we have included a schematic to help understand the graphs.

Figure 1

Figure 1: *nhs1b* is expressed in mesodermal cells during gastrulation. (A) Whole-mount *in situ* hybridisation with *nhs1b* or control probes, on embryos fixed at the 1-cell, shield (onset of gastrulation), 75% epiboly (mid-gastrulation) and 12-somite stages. (B-C) Whole mount *in situ* hybridisation with *nhs1b* and *sox17* probes and subsequent GFP immunostaining. (B) *nhs1b* is expressed in induced mesendodermal cells. N= 4 experiments. (C) No expression of *nhs1b* is observed in induced endodermal cells. N= 3 experiments. (D) Whole-mount double *in situ* hybridisation with *sox17* (red) and *nhs1b* (green) probes at 75% epiboly(end of gastrulation) and bud stages. *nhs1b* is not expressed in the *sox17* expressing fore-runner cells. Scale bar 100 μ m unless specified.

Figure 3

Figure 3: *nhs1b* knockdown reduces cell speed and persistence. (A) Representative image of mesodermal cell nuclei tracking, in a *Tg(tbx16:EGFP)* embryo expressing H2B-mCherry. (B-C) Instant cell speed and directionality ratio. Circles on the violin plot indicate mean per embryo. Likelihood ratio test of a linear mixed effects model with treatment as a fixed effect and embryos as a random effect against a model without the fixed effect. Adjusted p-values: (B) MO Control vs MO *Nhs1b*: 0,0100 *; MO Control vs MO-2 *Nhs1b*: 0,0131 *; (C) MO Control vs MO *Nhs1b*: 0,0301 *; MO Control vs MO-2 *Nhs1b*: 0,0301 *. (D) Directional autocorrelation. Likelihood ratio test of a non-linear mixed effects model (see methods) with treatment as a fixed effect and embryos as a random effect against a model without the fixed effect. Adjusted p-values: MO Control vs MO *Nhs1b*: 0,045 *; MO Control vs MO-2 *Nhs1b*: 0,0054 *. MO control (n=5), MO *Nhs1b* (n=8) and MO-2 *Nhs1b* (n=6) injected embryos. (E-F) Representative cell trajectories of mesodermal cells in MO control and MO *Nhs1b* injected embryos, colour-coded for cell migration persistence (directionality ratio).

Figure 6

Figure 6: Nhs1b regulates protrusion dynamics. (A) Protrusions of mesodermal cells injected with *Lifeact-mCherry* mRNAs and with a MO control or a MO Nhs1b, transplanted in the mesoderm of a non-labelled embryo. Selected time points showing the protrusion elongation. (B-C) Quantification of the lifetime and maximum length of protrusions. Likelihood ratio test of a linear mixed effects model with treatment as a fixed effect and cells as a random effect against a model without the fixed effect. Adjusted p-values: (B) MO Control vs MO Nhs1b: 0,0309*; MO Control vs MO-2 Nhs1b: 0,0011 ***; (C) MO Control vs MO Nhs1b: 0,0001***; MO Control vs MO-2 Nhs1b: 4,89E-06****. MO Control (n=5 embryos; n=12 cells), MO Nhs1b (n=6 embryos; n=14 cells) and MO-2 Nhs1b (n=7 embryos, 49 cells). (D) Protrusions of mesodermal cells injected with *Lifeact-mCherry* mRNAs and with *lacZ* (control) or *Nhs1b* mRNAs, transplanted in the mesoderm of a non-labelled embryo. Selected time points showing the protrusion elongation. (E-F) Quantification of the lifetime and maximum length of protrusions. Likelihood ratio test of a linear mixed effects model with treatment as a fixed effect and cells as a random effect against a model without the fixed effect. p-values: (E) Control vs Nhs1b: 0,0094**; (F) Control vs Nhs1b: 0,00012***. Control (n=5 embryos, 30 cells) or Nhs1b (n=6 embryos, 28 cells). Scale bars 20 μm .

Response to reviewer comments

We were very pleased to see that the reviewers found our revisions satisfactory. We sincerely thank them once again for their valuable comments, which have helped us improve the manuscript. We have now revised the manuscript according to their final suggestions. Below is a point-by-point response, with our replies highlighted in blue. The revised version of the manuscript clearly indicates all changes. We hope that the manuscript is now suitable for publication in *Communications Biology*.

Reviewers' comments:

Reviewer #1 (Remarks to the Author):

The authors have addressed all my concerns appropriately and I am now happy to support publication. Overall this is now a very well done study - congratulations!

Thank you !

Reviewer #2 (Remarks to the Author):

Summary of the manuscript

The revised manuscript by Escot et al. reports a role for *nhs1b*, a relatively understudied regulator of the Scar/WAVE complex, in mesodermal cell migration during zebrafish gastrulation. Using in vivo live imaging techniques, the authors show that *nhs1b* affects cell migration speed and directionality as well as actin and protrusion dynamics in mesodermal cells. Some of these effects are in contrast to previous reports on *NHSL1* function, suggesting that this actin regulator can act in a context-specific manner.

Overall impression of the work:

The authors have sufficiently addressed the reviewer comments. The manuscript is much improved especially with respect to additional information provided on statistical analyses used and confirmation and clarification of the mesoderm-specific expression of *nhs1b*. Minor suggestions are provided below to improve readability of the manuscript.

Specific comments

1. The addition of double fluorescent in situs to Fig. 1 are highly appreciated and strengthen the authors' conclusions that *nhs1b* is expressed in the mesoderm. I suggest displaying the panels in Fig. 1D in a green/magenta color scheme (as the authors did for all other figures) rather than green/red to improve accessibility.

Figure 1D is now displayed as green/magenta, as the other figures in the manuscript. Thanks for pointing this.

2. The dose-response analysis for *nhs1b* mRNA injection provided in the rebuttal letter is much appreciated. If possible, I encourage the authors to include these data in the manuscript (perhaps has

supplemental information) as it provides important controls for the rescue experiments and nicely demonstrates the authors' conclusions that *nhs1b* levels need to be finely tuned.

The dose response data, which were present in our initial response to reviewers, is now presented as Figure S2. Line 197, the following text was added: "Accordingly, overexpression phenotypes and rescue experiments were very sensitive to *nhs1b* mRNA doses (Fig. S2)."

3. The addition of axis labels and a schematic to Fig. 3E–F has improved readability of this figure. I encourage the authors to also include brief axis definitions to the figure legend (e.g., "y-axis represents distance along the animal-vegetal axis, x-axis represents distance along the dorsal-ventral axis").

The suggested sentence has been added to the figure legend.